# $f$-Divergence Policy Optimization in Fully Decentralized Cooperative MARL

**Kefan Su**                                           *sukefan@pku.edu.cn*
*School of Computer Science*
*Peking University*

**Zongqing Lu**[*]                                      *zongqing.lu@pku.edu.cn*
*School of Computer Science*
*Peking University*

Reviewed on OpenReview: *https://openreview.net/forum?id=Wj8yFjIpom*

## Abstract

Independent learning is a straightforward solution for fully decentralized learning in cooperative multi-agent reinforcement learning (MARL). The study of independent learning has a history of decades, and the representatives, such as independent Q-learning and independent PPO, can achieve good performances on several benchmarks. However, most independent learning algorithms lack convergence guarantees or theoretical support. In this paper, we propose a general formulation of independent policy optimization, $f$-divergence policy optimization. We hope that a more general policy optimization formulation will provide deeper insights into fully decentralized learning. We demonstrate the generality of this formulation and analyze its limitations. Based on this formulation, we further propose a novel independent learning algorithm, TVPO, which theoretically guarantees convergence. Empirically, we demonstrate that TVPO outperforms state-of-the-art fully decentralized learning methods on three popular cooperative MARL benchmarks, thereby verifying the efficacy of TVPO.

## 1 Introduction

Cooperative multi-agent reinforcement learning (MARL) has shown great potential in many areas, including power control (Zhang & Liang, 2020), autonomous vehicles (Han et al., 2022), and robot control (Sartoretti et al., 2019). The main framework for cooperative MARL is centralized training with decentralized execution (CTDE) (Kraemer & Banerjee, 2016), while the MARL community pays less attention to fully decentralized learning, also known as decentralized training with decentralized execution (DTDE). Fully decentralized learning remains significant in cooperative MARL due to its simplicity. From an applications perspective, fully decentralized learning is useful in various industrial applications where agents may belong to different parties, such as autonomous vehicles or robots. From a theoretical perspective, fully decentralized algorithms rely on less information during training and are therefore more general and worthy of further study.

For DTDE or fully decentralized settings, independent learning is a straightforward yet effective approach that enables agents to directly execute the same single-agent RL algorithm. Representative methods include independent Q-learning (IQL) (Tan, 1993) and independent actor-critic (IAC) (Foerster et al., 2018; Papoudakis et al., 2021). Recently, independent PPO (IPPO) (de Witt et al., 2020), which extends PPO (Schulman et al., 2017) to MARL, has shown good performance on several benchmarks. However, these independent learning algorithms are still troubled by the non-stationarity problem and lack convergence guarantees or theoretical support.

---

[*]Corresponding Author

Additionally, introducing constraints of the policy update or trust region into the policy optimization is a classic and effective paradigm for ensuring policy convergence guarantees in single-agent RL (Schulman et al., 2015; 2017; Nachum et al., 2017). Recently, DPO (Su & Lu, 2022b) has proposed a fully decentralized algorithm with a convergence guarantee by following this approach. However, DPO focuses on the KL-divergence and can be troubled by trivial policy updates in some cases due to its approximation bias in the optimization objective. Therefore, we hope that a more general policy optimization formulation can provide deeper insights into fully decentralized learning.

In this paper, we propose a general formulation of independent policy optimization, **$f$-divergence policy optimization**. We demonstrate the generality of this formulation for independent learning in cooperative MARL. We also analyze the policy iteration of this formulation and discuss its limitations using a two-player matrix game. Based on this formulation, we further propose a novel independent learning algorithm, **total variation policy optimization** (**TVPO**). To theoretically study the properties of TVPO and prove its convergence, we introduce a new set of value functions and policy iteration specifically designed for fully decentralized learning and prove the monotonicity of this policy iteration. The practical implementation of TVPO can be effectively realized through an adaptive coefficient, similar to PPO (Schulman et al., 2017).

Empirically, we verify our discussion regarding the limitations of $f$-divergence policy optimization in the two-player matrix game and demonstrate the joint policy may converge to a sub-optimal solution with different $f$-divergences. Moreover, we evaluate the performance of TVPO across three popular benchmarks of cooperative MARL including SMAC (Samvelyan et al., 2019), multi-agent MuJoCo (Peng et al., 2021) and SMACv2 (Ellis et al., 2023). We compare TVPO with four representative fully decentralized learning methods: IQL (Tan, 1993), IPPO (de Witt et al., 2020), I2Q (Jiang & Lu, 2022), and DPO (Su & Lu, 2022b). The empirical results show that TVPO outperforms these baselines in all evaluated tasks, thereby verifying the effectiveness of TVPO in fully decentralized cooperative MARL.

## 2 Related Work

**CTDE.** The popular framework to address cooperative multi-agent reinforcement learning (MARL) problems is centralized training with decentralized execution (CTDE) (Lowe et al., 2017; Foerster et al., 2018; Sunehag et al., 2018; Rashid et al., 2018; Iqbal & Sha, 2019; Wang et al., 2021a; Zhang et al., 2021; Su & Lu, 2022a; Wang et al., 2023a). CTDE successfully mitigates the challenge of non-stationarity through centralized training. This line of research can be categorized into two types: value decomposition algorithms (Sunehag et al., 2018; Rashid et al., 2018; Son et al., 2019; Yang et al., 2020; Wang et al., 2021a), where the optimum of the centralized Q-function aligns with the optima of the decentralized Q-functions, allowing the learning of the centralized Q-function to be factorized into the learning process of the decentralized Q-functions; and multi-agent actor-critic algorithms (Foerster et al., 2018; Iqbal & Sha, 2019; Wang et al., 2021b; Zhang et al., 2021; Su & Lu, 2022a; Wang et al., 2023a; Wen et al., 2022; Liu et al., 2023), which leverage a centralized Q-function to facilitate the learning of decentralized stochastic policies. HAPPO (Kuba et al., 2021) and MAPPO (Yu et al., 2021) extend the applicability of TRPO (Schulman et al., 2015) and PPO (Schulman et al., 2017), respectively, to the MARL setting through a centralized state value function. HASAC (Liu et al., 2023) combines the heterogeneous-agent decomposition with the entropy regularization in SAC. MAT (Wen et al., 2022) introduces Transformer and sequential modeling into the heterogeneous-agent decomposition. *Nevertheless, it is important to note that these approaches remain constrained by the CTDE paradigm and are therefore unsuitable for fully decentralized learning.*

**Fully Decentralized Learning.** There have recently been several different views on fully decentralized learning or decentralized learning. Some works study decentralized learning specifically with communication (Zhang et al., 2018b; Li et al., 2020) or parameter sharing (Terry et al., 2020). Both communication and parameter sharing involve exchanging information among agents (Terry et al., 2020). *However, in this paper, we consider fully decentralized learning in the strictest sense – with each agent independently learning its policy without being allowed to communicate or share parameters* as in Tampuu et al. (2015); Mao et al. (2022b); Wang et al. (2023c). Additionally, there are several studies (Zhan et al., 2023; Wang et al., 2023b; Mao & Başar, 2023) considering general-sum games in decentralized MARL, these studies focus on the episodic Markov game(Jin et al., 2021), which is non-cooperative and

assumes the reward function, transition probability, and policy are related to the time step. The objective of finding an equilibrium in this setting differs from the fully decentralized learning concerned in this paper. Independent learning (OroojlooyJadid & Hajinezhad, 2019) has been extensively studied in the field of cooperative multi-agent reinforcement learning (MARL) as a straightforward approach to fully decentralized learning. Representatives of this approach include independent Q-learning (IQL) (Tan, 1993; Tampuu et al., 2015), independent actor-critic (IAC) (Foerster et al., 2018; Papoudakis et al., 2021), and independent proximal policy optimization (IPPO) (de Witt et al., 2020). It should be noted that all these independent learning algorithms deviate from the stationary condition of the Markov decision process (MDP) and lack convergence guarantees, even though IQL and IPPO perform well in various benchmarks (Papoudakis et al., 2021). Recent studies have emerged with convergence guarantees in fully decentralized MARL, namely I2Q (Jiang & Lu, 2022) and DPO (Su & Lu, 2022b). I2Q introduces the concept of QSS-value (Edwards et al., 2020) into independent Q-learning, achieving convergence guarantees. However, its applicability is restricted to deterministic environments. On the other hand, a novel decentralized surrogate of the joint TRPO objective is proposed by DPO to ensure convergence. ***In terms of empirical performance, I2Q demonstrates superior performance compared to IQL, while DPO outperforms IPPO. Therefore, in our empirical studies, we comprehensively compare our TVPO with these two state-of-the-art methods.***

**Mirror Descent in RL.** Recently, mirror descent (Blair, 1985) and similar ideas have been applied in single-agent RL (Wang et al., 2019; Lan, 2023; Tomar et al., 2020; Yang et al., 2022; Vaswani et al., 2021) as well as in CTDE algorithms in MARL (Su & Lu, 2022a; Kuba et al., 2022; Liu et al., 2023) for theoretical guarantees. Mirror descent is a method associated with the Bregman divergence (Bregman, 1967). Although Bregman divergence is a general divergence class, KL-divergence has been most frequently used in previous mirror descent studies (Wang et al., 2019; Lan, 2023; Tomar et al., 2020; Yang et al., 2022; Vaswani et al., 2021). On the other hand, KL-divergence lies at the intersection of Bregman divergence and $f$-divergence and our analysis of KL-divergence indicates that it can become trapped in the sub-optimum even in a simple matrix game. Furthermore, Bregman divergence in mirror descent and $f$-divergence represent two distinct classes of divergences, and the latter can provide us with more useful properties for theoretical guarantees in fully decentralized learning. Extending mirror descent in fully decentralized learning with theoretical guarantees remains an open challenge and is beyond the scope of this paper.

## 3 Preliminaries

**Cooperative Markov Game.** The cooperative Markov Game serves as a general model for cooperative multi-agent reinforcement learning (MARL) (Chen et al., 2022; Zhang et al., 2018a; Matignon et al., 2012). It is a special case of the Markov Game (Littman, 1994) where the reward functions of all agents are identical. It is represented by the tuple $\mathcal{G} = \{S, A, P, I, N, r, \gamma\}$, where $N$ denotes the number of agents, and $I = \{1, 2, \cdots, N\}$ refers to the set of all agents. The state space is denoted as $S$, and the joint action space is denoted as $A = A_1 \times A_2 \times \cdots \times A_N$, where $a^i$ represents the individual action space for agent $i$. The transition function $P(s'|s, \boldsymbol{a}) : S \times A \times S \to [0, 1]$ defines the probability of transitioning from state $s$ to $s'$ given a joint action $\boldsymbol{a}$. The discount factor is denoted as $\gamma \in [0, 1)$, and the reward function $r(s, \boldsymbol{a}) : S \times A \to [-r_{\max}, r_{\max}]$ assigns rewards to state $s$ and joint action $\boldsymbol{a}$, with $r_{\max}$ serving as the upper bound of the reward function. The objective of cooperative Markov Game is to maximize $J(\boldsymbol{\pi}) = \mathbb{E}_{\boldsymbol{\pi}} \left[ \sum_{t=0} \gamma^t r(s_t, \boldsymbol{a}_t) \right]$. Thus, the optimal joint policy $\boldsymbol{\pi}^* = \arg \max_{\boldsymbol{\pi}} J(\boldsymbol{\pi})$ needs to be determined. In fully decentralized learning, each agent independently learns an individual policy denoted as $\pi^i(a^i|s)$. The joint policy $\boldsymbol{\pi}$ of all agents can be represented as the product of each individual policy $\pi^i$.

Additionally, the V-function and Q-function of the joint policy $\boldsymbol{\pi}$ can be defined as follows:

$$V^{\boldsymbol{\pi}}(s) = \mathbb{E}_{\boldsymbol{a} \sim \boldsymbol{\pi}} \left[ Q^{\boldsymbol{\pi}}(s, \boldsymbol{a}) \right], \tag{1}$$

$$Q^{\boldsymbol{\pi}}(s, \boldsymbol{a}) = r(s, \boldsymbol{a}) + \gamma \mathbb{E}_{s' \sim P(\cdot|s, \boldsymbol{a})} \left[ V^{\boldsymbol{\pi}}(s') \right]. \tag{2}$$

**Fully Decentralized Critic.** The concept of the critic in fully decentralized learning has been explored in previous studies (Peshkin et al., 2000; Lyu & Xiao, 2021; Su & Lu, 2022b). To facilitate further discussion, we provide some formulations and deductions regarding the fully decentralized critic.

In fully decentralized learning, each agent learns independently through its own interactions with the environment. Consequently, the Q-function for each agent $i$ can be described by the following formula:

$$Q_{\pi^{-i}}^{\pi^i}(s, a^i) = r_{\pi^{-i}}(s, a^i) + \gamma \mathbb{E}_{a^{-i} \sim \pi^{-i}(\cdot|s')}[Q_{\pi^{-i}}^{\pi^i}(s', a^{i'})], \tag{3}$$

where $r_{\pi^{-i}}(s, a^i) = \mathbb{E}_{a^{-i} \sim \pi^{-i}(\cdot|s')}[r(s, a^i, a^{-i})]$, and $\pi^{-i}$ and $a^{-i}$ respectively denote the joint policy and joint action of all agents expect agent $i$. It can be shown that $Q_{\pi^{-i}}^{\pi^i}(s, a^i) = \mathbb{E}_{a^{-i} \sim \pi^{-i}(\cdot|s')}[Q^{\boldsymbol{\pi}}(s, a^i, a^{-i})]$. For simplicity, in the following, we use $Q_i^{\boldsymbol{\pi}}$ to denote $Q_{\pi^{-i}}^{\pi^i}$ given a joint policy $\boldsymbol{\pi}$, if there is no confusion.

**Independent Learning.** Independent learning is a straightforward method to solve cooperative MARL problems, which makes each agent learn through the same single-agent RL algorithm, such as IQL (Tan, 1993), IAC (Foerster et al., 2018), and IPPO (de Witt et al., 2020). Though independent learning faces the non-stationarity problem, it still has the advantage of absorbing the benefit of single-agent RL. The policy iteration $\pi_{\text{new}} = \arg\max_\pi \sum_a \pi(a|s) Q^{\pi_{\text{old}}}(s, a)$ is fundamental in single-agent RL, which ensures that $\pi_{\text{new}}$ improves monotonically over $\pi_{\text{old}}$ and guarantees the convergence. We draw inspiration from policy iteration in single-agent RL, introduce a general formulation of independent policy optimization, and try to find an independent learning algorithm that can guarantee convergence in cooperative MARL.

# 4 A General Formulation for Independent Policy Optimization

Given the condition of fully decentralized learning in cooperative MARL, we first propose a general formulation of independent policy optimization, $f$-divergence policy optimization, and discuss its generality and limitation. Then, based on this formulation, we propose total variation policy optimization (TVPO), prove the convergence of TVPO in fully decentralized learning, and provide a practical algorithm.

|  | | $u_B^0$ | $u_B^1$ |
|---|---|---|---|
| Bob
Alice | | $q_t$ | $1 - q_t$ |
| $u_A^0$ | $p_t$ | $a$ | $b$ |
| $u_A^1$ | $1 - p_t$ | $c$ | $d$ |

Table 1: The two-player matrix game for Alice and Bob with policies after the number $t$ of policy iterations. Alice will take action $u_A^0$ with probability $p_t$ and take action $u_A^1$ with probability $1 - p_t$; Bob will take action $u_B^0$ with probability $q_t$ and take action $u_B^1$ with probability $1 - q_t$.

Before diving into the discussion, we need to introduce a simple two-player matrix game for later use. In this matrix game, the two agents, Alice and Bob, each have two actions, denoted as $\{u_A^0, u_A^1\}$ for Alice and $\{u_B^0, u_B^1\}$ for Bob. Each episode of this matrix game consists of only one step. The rewards for the joint actions $(u_A^0, u_B^0)$, $(u_A^0, u_B^1)$, $(u_A^1, u_B^0)$ and $(u_A^1, u_B^1)$ are $a$, $b$, $c$, and $d$ respectively. The policies of Alice and Bob can be described with $p_t$ and $q_t$ as that Alice will take action $u_A^0$ with probability $p_t$ and Bob will take action $u_B^0$ with probability $q_t$, where $t$ represents the number of policy iterations. The full information of this matrix game is illustrated in Table 1.

## 4.1 $f$-Divergence Policy Optimization

The $f$-divergence policy optimization is formulated as follows:

**Definition 4.1.** Given any fixed $s$ and $\pi_{\text{old}}^i$

$$\pi_{\text{new}}^i = \arg\max_{\pi^i} \sum_{a^i} \pi^i(a^i|s) Q_i^{\boldsymbol{\pi}_{\text{old}}}(s, a^i) - \omega D_f\left(\pi^i(\cdot|s) \| \pi_{\text{old}}^i(\cdot|s)\right), \tag{4}$$

where $D_f(p\|q) \triangleq \sum_i q_i f\left(\frac{p_i}{q_i}\right)$ is $f$-divergence (Ali & Silvey, 1966) and according to the definition of $f$-divergence, $f : [0, \infty) \to (-\infty, +\infty]$ is convex and $f(1) = 0$.

This formulation contains an additional term $D_f\left(\pi^i(\cdot|s)\|\pi_{\text{old}}^i(\cdot|s)\right)$, which describes the distance between $\pi^i$ and $\pi_{\text{old}}^i$.

There are several studies considering the distance between $\boldsymbol{\pi}_{\text{old}}$ and $\boldsymbol{\pi}_{\text{new}}$. The trust region in TRPO (Schulman et al., 2015) and PPO (Schulman et al., 2017) is actually KL-divergence between $\boldsymbol{\pi}_{\text{old}}$ and $\boldsymbol{\pi}_{\text{new}}$,

while Nachum et al. (2017) extend entropy regularization to a more general formulation with KL-divergence. Unlike these studies that just use KL-divergence as the distance measure, we would like to discuss a more general formulation. So we use $f$-divergence, which is widely used for describing the distance between two distributions. Also, KL-divergence is a special case of $f$-divergence with $f(x) = x \log x$ and we have many other choices for $f$-divergence, such as $f(x) = \frac{|x-1|}{2}$ corresponding to total variation distance $D_f(p\|q) = \frac{1}{2} \sum_i |p_i - q_i|$ and $f(x) = (1 - \sqrt{x})^2$ corresponding to Hellinger distance $D_f(p\|q) = \sqrt{\sum_i (\sqrt{p_i} - \sqrt{q_i})^2}$.

To further discuss $f$-divergence policy optimization, we need to find the solution to the optimization objective (4) and we have the following lemma.

**Lemma 4.2.** *Given a fixed function $f$ and the corresponding $f$-divergence $D_f$, let $g(x) = (f')^{-1}(x)$, then the solution to Equation (4) is*

$$\pi_{\text{new}}^i(a^i|s) = \max\{\pi_{\text{old}}^i(a^i|s)g\left(\frac{\lambda_s + Q_i^{\boldsymbol{\pi}_{\text{old}}}(s, a^i)}{\omega}\right), 0\}, \tag{5}$$

*where $\lambda_s$ satisfies*

$$\sum_{a^i} \max\{\pi_{\text{old}}^i(a^i|s)g\left(\frac{\lambda_s + Q_i^{\boldsymbol{\pi}_{\text{old}}}(s, a^i)}{\omega}\right), 0\} = 1.$$

This proof is included in Appendix A.1 and follows Yang et al. (2019).

We use the two-player matrix game between Alice and Bob (*i.e.*, Table 1) to discuss the limitation of $f$-divergence policy optimization. As for the policy iteration in the matrix game, we have the following proposition.

**Proposition 4.3.** *Suppose $g(x) \geq 0$ and let $M = b + c - a - d$, $\hat{p} = \frac{c-d}{M}$, and $\hat{q} = \frac{b-d}{M}$. If the payoff matrix of the two-player matrix game satisfies $M > 0$, and Alice and Bob update their policies with*

$$\pi_{t+1}^i = \arg \max_{\pi^i} \sum_{a^i} \pi^i(a^i|s)Q_i^{\boldsymbol{\pi}_t}(s, a^i) - \omega D_f\left(\pi^i(\cdot|s)\|\pi_t^i(\cdot|s)\right), \tag{6}$$

*then we have (1) $p_t \geq \hat{p} \Rightarrow q_{t+1} \leq q_t$; (2) $p_t \leq \hat{p} \Rightarrow q_{t+1} \geq q_t$; (3) $q_t \geq \hat{q} \Rightarrow p_{t+1} \leq p_t$; (4) $q_t \leq \hat{q} \Rightarrow p_{t+1} \geq p_t$.*

The proof is included in Appendix A.2. With Proposition 4.3, we can build a case where the joint policy sequence can only converge to the sub-optimum. We assume the matrix game satisfies the condition $b > c > \max\{a, d\}$, then the optimal joint policy is $(p_t, q_t) = (1, 0)$ corresponding to the joint action $(u_A^0, u_B^1)$ and reward $b$. Moreover, the condition $b > c > \max\{a, d\}$ also means $\hat{p} \in (0, 1)$ and $\hat{q} \in (0, 1)$. If at iteration $t$, the condition $q_t > \hat{q}$, $p_t < \hat{p}$ is satisfied, then $q_{t+1} > q_t > \hat{q}$, $p_{t+1} < p_t < \hat{p}$. By induction, we know that $\forall t' \geq t$, $q_{t'+1} > q_{t'} > \hat{q}$, $p_{t'+1} < p_{t'} < \hat{p}$. As the sequence $\{p_t\}$ and $\{q_t\}$ are both bounded in the interval $[0, 1]$, we know the sequence $\{p_t\}$ and $\{q_t\}$ will converge to $p^*$ and $q^*$. As for $p^*$ and $q^*$, we have the following corollary.

**Corollary 4.4.** *If at iteration $t$, the condition $q_t > \hat{q}$, $p_t < \hat{p}$ is satisfied, then the sequence $\{p_t\}$ and $\{q_t\}$ will converge to $p^* = 0$ and $q^* = 1$ respectively.*

The proof is included in Appendix A.3. Corollary 4.4 tells us if once $q_t > \hat{q}$, $p_t < \hat{p}$, then the joint policy converges to the sub-optimal solution $(p^*, q^*) = (0, 1)$ corresponding to the joint action $(u_A^1, u_B^0)$ and reward $c$. So if the initial policy $p_0$ and $q_0$ satisfies the condition $q_0 > \hat{q}$, $p_0 < \hat{p}$, then the joint policy converges to the sub-optimal policy. We further illustrate this in the experiment.

## 4.2 Total Variation Policy Optimization

The $f$-divergence formulation (4) can be trapped in the sub-optimal joint policy even in a simple two-player matrix game. This shows the upper bound of $f$-divergence policy optimization, so we should not expect such a policy iteration could obtain the optimal joint policy in fully decentralized learning in all MDPs. Fortunately, we have found an algorithm that accords with the $f$-divergence formulation and has

the convergence guarantee. This algorithm uses the total variation distance for $f$-divergence, so we call it total variation policy optimization (TVPO). The convergence guarantee of TVPO shows the potential of the $f$-divergence formulation.

Before we introduce TVPO and prove its convergence, we need some definitions and lemmas. We use $D_{\text{TV}}(p\|q) \triangleq \frac{1}{2}\sum_i |p_i - q_i|$ to represent the total variation distance. We define a new V-function $V_{\boldsymbol{\rho}}^{\boldsymbol{\pi}}(s)$ and a new Q-function $Q_{\boldsymbol{\rho}}^{\boldsymbol{\pi}}(s, a^i, a^{-i})$ given joint polices $\boldsymbol{\pi}$ and $\boldsymbol{\rho}$ as follows:

**Definition 4.5.**

$$V_{\boldsymbol{\rho}}^{\boldsymbol{\pi}}(s) = \frac{1}{N}\sum_i \sum_{a^i} \pi^i(a^i|s) \sum_{a^{-i}} \rho^{-i}(a^{-i}|s) Q_{\boldsymbol{\rho}}^{\boldsymbol{\pi}}(s, a^i, a^{-i}) - \omega D_f\left(\pi^i(\cdot|s)\|\rho^i(\cdot|s)\right), \tag{7}$$

$$Q_{\boldsymbol{\rho}}^{\boldsymbol{\pi}}(s, a^i, a^{-i}) = r(s, a^i, a^{-i}) + \gamma\mathbb{E}\left[V_{\boldsymbol{\rho}}^{\boldsymbol{\pi}}(s')\right]. \tag{8}$$

As the definition (7) is a fixed-point equation, we need to prove that this definition is well-defined. So we define an operator $\Gamma_{\boldsymbol{\rho}}^{\boldsymbol{\pi}}$ as follows:

$$\Gamma_{\boldsymbol{\rho}}^{\boldsymbol{\pi}} V(s) = \frac{1}{N}\sum_i \sum_{a^i} \pi^i(a^i|s) \sum_{a^{-i}} \rho^{-i}(a^{-i}|s)\left(r(s, \boldsymbol{a}) + \gamma\mathbb{E}\left[V(s')\right]\right) - \omega D_f\left(\pi^i(\cdot|s)\|\rho^i(\cdot|s)\right). \tag{9}$$

Then for any value function $V_1$ and $V_2$, we have

$$\left\|\Gamma_{\boldsymbol{\rho}}^{\boldsymbol{\pi}} V_1(s) - \Gamma_{\boldsymbol{\rho}}^{\boldsymbol{\pi}} V_2(s)\right\|_{\infty} = \gamma\left\|\frac{1}{N}\sum_i \sum_{a^i} \pi^i(a^i|s) \sum_{a^{-i}} \rho^{-i}(a^{-i}|s)\left(\mathbb{E}\left[V_1(s')\right] - \mathbb{E}\left[V_2(s')\right]\right)\right\|_{\infty}$$

$$\leq \gamma\|V_1(s) - V_2(s)\|_{\infty}.$$

So the operator $\Gamma_{\boldsymbol{\rho}}^{\boldsymbol{\pi}}$ is a $\gamma$-contraction, which means $V_{\boldsymbol{\rho}}^{\boldsymbol{\pi}}(s)$ is the unique fixed-point of (7) and the definition (7) is well-defined.

To apply total variation distance to independent policy optimization, we have the following lemma.

**Lemma 4.6.** *Suppose $\boldsymbol{\pi}_{\text{new}}$, $\boldsymbol{\pi}_{\text{old}}$, and $\boldsymbol{\pi}$ are three joint policies. Let $L = \frac{2r_{\max}}{1-\gamma}$, then for any state $s$, we have*

$$\sum_{\boldsymbol{a}} \boldsymbol{\pi}_{\text{new}}(\boldsymbol{a}|s) Q^{\boldsymbol{\pi}}(s, \boldsymbol{a}) \geq \frac{1}{N}\sum_{i=1}^{N} \sum_{a^i} \pi_{\text{new}}^i(a^i|s) \sum_{a^{-i}} \pi_{\text{old}}^{-i}(a^{-i}|s) Q^{\boldsymbol{\pi}}(s, a^i, a^{-i})$$

$$- \frac{(N-1)L}{N}\sum_{i=1}^{N} D_{\text{TV}}\left(\pi_{\text{new}}^i(\cdot|s)\|\pi_{\text{old}}^i(\cdot|s)\right). \tag{10}$$

The proof is included in Appendix A.4. Lemma 4.6 is a critical bridge between normal value function $V^{\boldsymbol{\pi}}$ and our new value function $V_{\boldsymbol{\rho}}^{\boldsymbol{\pi}}$, and we can witness its effect in our later discussion. Moreover, we also know that $V_{\boldsymbol{\pi}}^{\boldsymbol{\pi}} = V^{\boldsymbol{\pi}}$ and $Q_{\boldsymbol{\pi}}^{\boldsymbol{\pi}} = Q^{\boldsymbol{\pi}}$.

We can also realize the monotonic improvement with a fully decentralized optimization objective via the following proposition.

**Proposition 4.7.** *Given a fixed joint policy $\boldsymbol{\rho}$ and an old joint policy $\boldsymbol{\pi}_{\text{old}}$, if all the agents update their policies according to*

$$\pi_{\text{new}}^i = \arg\max_{\pi^i} \sum_{a^i} \pi^i(a^i|s) \sum_{a^{-i}} \rho^{-i}(a^{-i}|s) Q_{\boldsymbol{\rho}}^{\boldsymbol{\pi}_{\text{old}}}(s, \boldsymbol{a}) - \omega D_f\left(\pi^i(\cdot|s)\|\rho^i(\cdot|s)\right), \tag{11}$$

*then we have $V_{\boldsymbol{\rho}}^{\boldsymbol{\pi}_{\text{old}}}(s) \leq V_{\boldsymbol{\rho}}^{\boldsymbol{\pi}_{\text{new}}}(s)$, $Q_{\boldsymbol{\rho}}^{\boldsymbol{\pi}_{\text{old}}}(s, \boldsymbol{a}) \leq Q_{\boldsymbol{\rho}}^{\boldsymbol{\pi}_{\text{new}}}(s, \boldsymbol{a})$, $\forall s \in S, \boldsymbol{a} \in A$.*

The proof is included in Appendix A.5. According to (11), by taking $\boldsymbol{\pi}_{\text{old}} = \boldsymbol{\rho} = \boldsymbol{\pi}_t$ and $\boldsymbol{\pi}_{\text{new}} = \boldsymbol{\pi}_{t+1}$, we can design a policy iteration as follows:

$$\pi_{t+1}^i = \arg\max_{\pi^i} \sum_{a^i} \pi^i(a^i|s) \sum_{a^{-i}} \pi_t^{-i}(a^{-i}|s) Q^{\boldsymbol{\pi}_t}(s, a^i, a^{-i}) - \omega D_f\left(\pi^i(\cdot|s)\|\pi_t^i(\cdot|s)\right). \tag{12}$$

This policy iteration resolves the $f$-divergence formulation (4). According to Proposition 4.7, we know the joint policy sequence $\{\boldsymbol{\pi}_t\}$ has the property $V_{\boldsymbol{\pi}_t}^{\boldsymbol{\pi}_{t+1}}(s) \geq V_{\boldsymbol{\pi}_t}^{\boldsymbol{\pi}_t}(s) = V^{\boldsymbol{\pi}_t}(s)$. By taking $D_f = D_{\text{TV}}$ and $\omega = \frac{(N-1)L}{N}$, we can combine these results with Lemma 4.6 to obtain the convergence guarantee.

**Theorem 4.8.** *Let $\omega = \frac{(N-1)L}{N}$. If all agents update their policies according to*

$$\pi_{t+1}^i = \arg\max_{\pi^i} \sum_{a^i} \pi^i(a^i|s) \sum_{a^{-i}} \pi_t^{-i}(a^{-i}|s) Q^{\boldsymbol{\pi}_t}(s, a^i, a^{-i}) - \omega D_{\text{TV}}\left(\pi^i(\cdot|s) \| \pi_t^i(\cdot|s)\right)$$

$$= \arg\max_{\pi^i} \sum_{a^i} \pi^i(a^i|s) Q_i^{\boldsymbol{\pi}_t}(s, a^i) - \omega D_{\text{TV}}\left(\pi^i(\cdot|s) \| \pi_t^i(\cdot|s)\right), \tag{13}$$

*then we have $V_{\boldsymbol{\pi}_t}^{\boldsymbol{\pi}_{t+1}}(s) \geq V^{\boldsymbol{\pi}_t}(s) \geq V_{\boldsymbol{\pi}_{t-1}}^{\boldsymbol{\pi}_t}(s) \geq V^{\boldsymbol{\pi}_{t-1}}(s)$. Moreover, the sequence $\{V^{\boldsymbol{\pi}_t}\}$ and $\{\boldsymbol{\pi}_t\}$ converge to $V^*$ and $\boldsymbol{\pi}_*$ respectively, which satisfy the fixed-point equation,*

$$\pi_*^i = \arg\max_{\pi^i} \sum_{a^i} \pi^i(a^i|s) \sum_{a^{-i}} \pi_*^{-i}(a^{-i}|s) \left(r(s, a^i, a^{-i}) + \gamma \mathbb{E}\left[V^*(s')\right]\right)$$

$$- \omega D_{\text{TV}}\left(\pi^i(\cdot|s) \| \pi_*^i(\cdot|s)\right).$$

The proof is included in Appendix A.6.

We further discuss the coefficient $\omega$. Intuitively, if $\omega$ is too large, then the policy will not be updated by (13), *i.e.*, (13) only has a trivial solution $\pi_{t+1}^i = \pi_t^i$. A similar conclusion has been mentioned in Schulman et al. (2015). For the total variation distance case, the threshold value of $\omega$ is $M = \frac{2r_{\max}}{1-\gamma} = 2\|Q\|_\infty$. For any $\tilde{\omega} > M$, we can show that (13) only has a trivial solution $\pi_{t+1}^i = \pi_t^i$. From the property of (13), we have

$$\left\langle \pi_{t+1}^i, Q^{\boldsymbol{\pi}_t} \right\rangle - \frac{\tilde{\omega}}{2} \left\| \pi_{t+1}^i - \pi_t^i \right\|_1 \geq \left\langle \pi_t^i, Q^{\boldsymbol{\pi}_t} \right\rangle \Rightarrow \left\langle \pi_{t+1}^i - \pi_t^i, Q^{\boldsymbol{\pi}_t} \right\rangle - \frac{\tilde{\omega}}{2} \left\| \pi_{t+1}^i - \pi_t^i \right\|_1 \geq 0$$

$$\Rightarrow \left( \|Q^{\boldsymbol{\pi}_t}\|_\infty - \frac{\tilde{\omega}}{2} \right) \left\| \pi_{t+1}^i - \pi_t^i \right\|_1 \geq 0. \tag{14}$$

The step (14) is from the inequality $\left\langle \pi_{t+1}^i - \pi_t^i, Q^{\boldsymbol{\pi}_t} \right\rangle \leq \left\| \pi_{t+1}^i - \pi_t^i \right\|_1 \|Q^{\boldsymbol{\pi}_t}\|_\infty$. Thus, the condition $\tilde{\omega} > M$ indicates $\left\| \pi_{t+1}^i - \pi_t^i \right\|_1 = 0$ which results in the trivial solution. Our choice of $\omega = \frac{(N-1)L}{N}$ has two critical properties. On the one hand, if $N = 1$, then $\omega = 0$ and (13) degenerates to a single-agent policy update. On the other hand, $\omega < M$ indicates the possibility of the non-trivial update $\pi_{t+1}^i \neq \pi_t^i$. We can show the non-trivial update of (13) in a two-player matrix game in Table 1 with both theoretical and empirical results. More details about the non-trivial update are included in Appendix F.4 and Section 5.2.

**Remark.** The policy optimization objective of TVPO is (13). An important property of (13) is that it can be optimized individually and independently by each agent and the joint policy converges according to Theorem 4.8. Although (13) is similar to the surrogate of DPO (Su & Lu, 2022b), there are two main differences between TVPO and DPO. The first difference is that from the property $D_{\text{TV}}^2(p\|q) \leq D_{\text{KL}}(p\|q)$, the bound $D_{\text{TV}}$ of TVPO is tighter than $\sqrt{D_{\text{KL}}}$ in DPO. The second difference is that TVPO obtains the convergence guarantee through policy iteration while DPO obtains the convergence guarantee through the surrogate of joint TRPO objective. A tighter bound means the iteration is less likely to be influenced by the trivial update. We also investigated their empirical performance in the experiments. More details of the discussion about the difference between TVPO and DPO are included in Section 5.2.

## 4.3 The Practical Algorithm of TVPO

Practically, if we use the objective (13) directly, then the large coefficient $\omega$ will greatly limit the step size of the policy update, and the algorithm will not work (Schulman et al., 2015). So we follow previous studies such as PPO (Schulman et al., 2017) to use an adaptive coefficient $\beta^i$ to replace $\omega$, then the policy optimization objective can be rewritten as

$$\pi_{t+1}^i = \arg\max_{\pi^i} \sum_{a^i} \pi^i(a^i|s) A_i^{\boldsymbol{\pi}_t}(s, a^i) - \beta^i D_{\text{TV}}\left(\pi^i(\cdot|s) \| \pi_t^i(\cdot|s)\right), \tag{15}$$

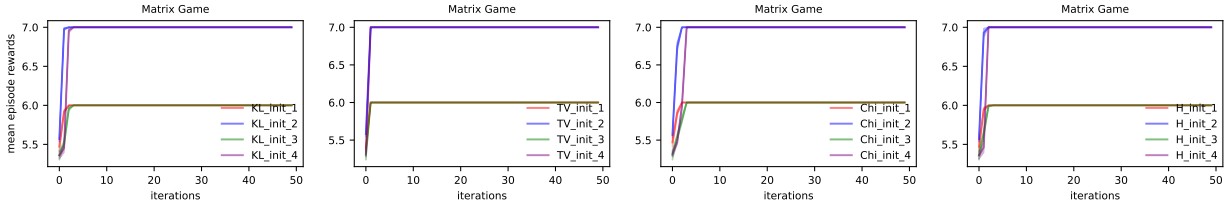

Figure 1: Learning curves of KL-iteration, TV-iteration, $\chi^2$-iteration, and H-iteration over four different sets of initialization in the matrix game (Table 1).

where $A_i^{\boldsymbol{\pi}_t}(s, a^i) = Q_i^{\boldsymbol{\pi}_t}(s, a^i) - \mathbb{E}_{\pi_t^i}\left[Q_i^{\boldsymbol{\pi}_t}(s, a^i)\right] = Q_i^{\boldsymbol{\pi}_t}(s, a^i) - V^{\boldsymbol{\pi}_t}(s)$. Here we use the baseline $V^{\boldsymbol{\pi}_t}(s)$ to reduce the variance in training.

The update rule of $\beta^i$ follows the practice of PPO. We can choose a hyperparameter $d$, which means we expect the total variation distance should be around $d$. Then we can update $\beta^i$ according to the value of $D_{\mathrm{TV}}\left(\pi_{t+1}^i(\cdot|s)\|\pi_t^i(\cdot|s)\right)$ in training as follows:

$$
\begin{aligned}
&\text{if } D_{\mathrm{TV}}\left(\pi_{t+1}^i(\cdot|s)\|\pi_t^i(\cdot|s)\right) > d * \delta, \text{then } \beta^i \leftarrow \beta^i \times \alpha \\
&\text{if } D_{\mathrm{TV}}\left(\pi_{t+1}^i(\cdot|s)\|\pi_t^i(\cdot|s)\right) < d/\delta, \text{then } \beta^i \leftarrow \beta^i/\alpha,
\end{aligned}
\tag{16}
$$

where $\delta$ and $\alpha$ are two constants and we choose $\delta = 1.5$ and $\alpha = 2$ like the choice of PPO.

For the critic, since the policy update needs to calculate $A_i^{\boldsymbol{\pi}_t}(s, a^i) = \mathbb{E}_{\pi_t^{-i}}[r(s, a^i, a^{-i}) + \gamma V^{\boldsymbol{\pi}_t}(s') - V^{\boldsymbol{\pi}_t}(s)]$, we take an individual state value function $V^i(s)$ as the critic for each agent $i$ and approximate $A_i^{\boldsymbol{\pi}_t}(s, a^i)$ with $\hat{A}_i = r + \gamma V^i(s') - V^i(s)$. The critic is updated as follows:

$$
\mathcal{L}_{\mathrm{critic}}^i = \mathbb{E}\left[(V^i(s) - y_i)^2\right],
\tag{17}
$$

where $y_i = r + \gamma V^i(s')$ or other target values.

When facing continuous action space, we usually use Gaussian distribution as the policy. However, there is no closed-form solution for total variation distance between two Gaussian distributions, to the best of our knowledge. To avoid optimization difficulties, we replace total variation distance with Hellinger distance $D_{\mathrm{H}}(p\|q) = \sqrt{\sum_i(\sqrt{p_i} - \sqrt{q_i})^2}$ in the environment with continuous action space, since there is a closed-form solution for Hellinger distance between two Gaussian distributions. Moreover, Hellinger distance has a critical property related to total variation distance that $D_{\mathrm{TV}}(p\|q) \leq D_{\mathrm{H}}(p\|q)$ and the proof is included in Appendix A.7.

With this property, we can replace $D_{\mathrm{TV}}$ with $D_{\mathrm{H}}$ in Lemma 4.6 and Theorem 4.8, while we can still obtain the same convergence guarantee. Thus, for the continuous action space, we use the following policy optimization objective:

$$
\pi_{t+1}^i = \arg\max_{\pi^i} \sum_{a^i} \pi^i(a^i|s)A_i^{\boldsymbol{\pi}_t}(s, a^i) - \beta^i D_{\mathrm{H}}\left(\pi^i(\cdot|s)\|\pi_t^i(\cdot|s)\right).
\tag{18}
$$

The practical algorithm of TVPO is summarized in Algorithm 1 in Appendix C.

## 5 Experiments

The experiments contain four main parts. The first part is to verify the limitation of $f$-divergence policy optimization as we have discussed in Section 4.1 through the matrix game. The second part is to compare TVPO with DPO in a matrix game. The third part is to evaluate the performance of TVPO in three popular cooperative MARL benchmarks including SMAC (Samvelyan et al., 2019), multi-agent MuJoCo (Peng et al., 2021) and SMACv2 (Ellis et al., 2023), compared with state-of-the-art fully decentralized algorithms. The last part is the ablation study about the hyperparameters $d$, $\alpha$ and $\beta$. All learning curves correspond to five different random seeds and the shaded area corresponds to the 95% confidence interval. To ensure reproducibility, our codes are included in the supplementary material and will be open source upon acceptance. Due to the space limit, additional experiments are included in Appendix E.

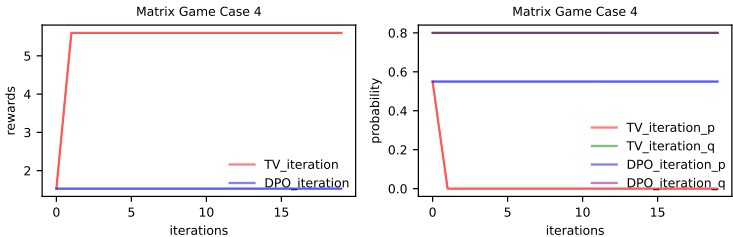

Figure 2: Learning curves of the iteration (13) and the DPO iteration in the matrix game $(a, b, c, d) = (-4, 7, 6, 4)$, where x-axis is iteration steps. The first and second figures show the expectation $J(\boldsymbol{\pi}_t)$ and the policies $p$ and $q$ of two iterations in the matrix game case 4 respectively, where $J(\boldsymbol{\pi}_t)$ is calculated by the joint policy $\boldsymbol{\pi}_t = (p_t, q_t)$ and the payoff matrix.

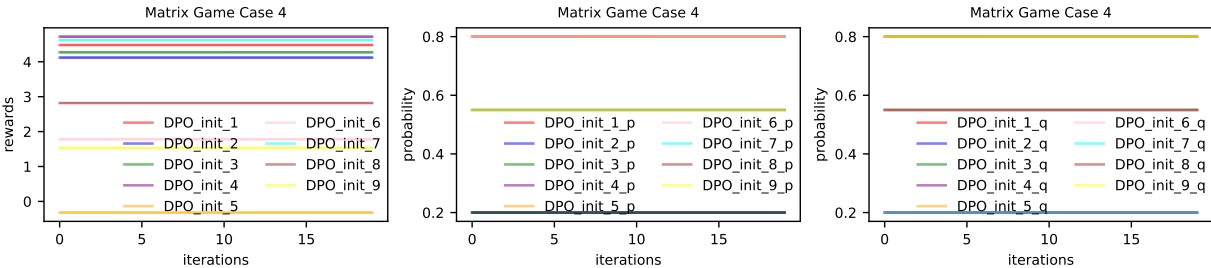

Figure 3: Learning curves of the DPO iteration with different initial policies in the matrix game $(a, b, c, d) = (-4, 7, 6, 4)$, where x-axis is iteration steps. The three figures show the expectation $J(\boldsymbol{\pi}_t)$, the policies $p$ and $q$ of nine different initial policies in the matrix game case 4 respectively, where $J(\boldsymbol{\pi}_t)$ is calculated by the joint policy $\boldsymbol{\pi}_t = (p_t, q_t)$ and the payoff matrix.

## 5.1 Verification in Matrix Game

In this section, we choose $a = 5$, $b = 7$, $c = 6$, $d = 4$ for the matrix game, which satisfies the condition $b > c > \max\{a, d\}$ as mentioned in Section 4.1. We use four different specific $f$-divergences: KL-divergence, total variation distance, $\chi^2$-distance, and Hellinger distance to build four different iterations of (4). We call these four iterations as KL-iteration, TV-iteration, $\chi^2$-iteration, and H-iteration respectively. We test these iterations over four sets of initialization: init_1 $(p_0, q_0) = (0.4, 0.8)$; init_2 $(p_0, q_0) = (0.6, 0.6)$; init_3 $(p_0, q_0) = (0.49, 0.76)$; init_4 $(p_0, q_0) = (0.51, 0.74)$. For the matrix game, we can calculate that $(\hat{p}, \hat{q}) = (0.5, 0.75)$ as defined in Proposition 4.3. From the discussion in Section 4.1 we know that init_1 and init_3 satisfy the condition $p_0 < \hat{p}$, $q_0 > \hat{q}$, which means

Table 2: The policy update types of DPO iteration with different initial policies in the matrix game $(a, b, c, d) = (-4, 7, 6, 4)$. $T$ represents the trivial policy update and $NT$ represents the non-trivial policy update.

| $q_0$ $p_0$ | 0.2 | 0.55 | 0.8 |
|---|---|---|---|
| 0.2 | $T$ | $T$ | $T$ |
| 0.55 | $T$ | $T$ | $T$ |
| 0.8 | $T$ | $T$ | $T$ |

the converged policy should be the sub-optimal policy $(p^*, q^*) = (0, 1)$ with reward $c = 6$, and init_2 and init_4 satisfy the condition $p_0 > \hat{p}$, $q_0 < \hat{q}$, which means the converged policy should be the optimal policy $(p^*, q^*) = (1, 0)$ with reward $b = 7$. The empirical results are illustrated in Figure 1. We can find that the empirical results agree with our theoretical derivation for all four iterations over the four sets of initialization. The learning curves of the policy $p$ and $q$ are included in Figure 9 in Appendix E. These empirical results corroborate our discussion about the limitation of $f$-divergence formulation.

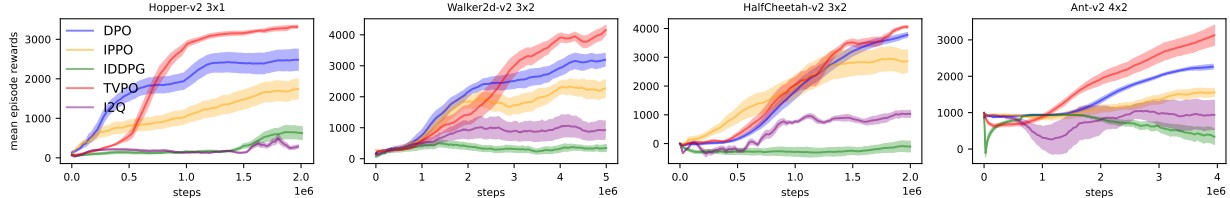

Figure 4: Learning curves of TVPO compared with IQL, IPPO, I2Q, and DPO on the maps 2s3z, 3s5z, 8m, MMM2 and 27m_vs_30m in SMAC.

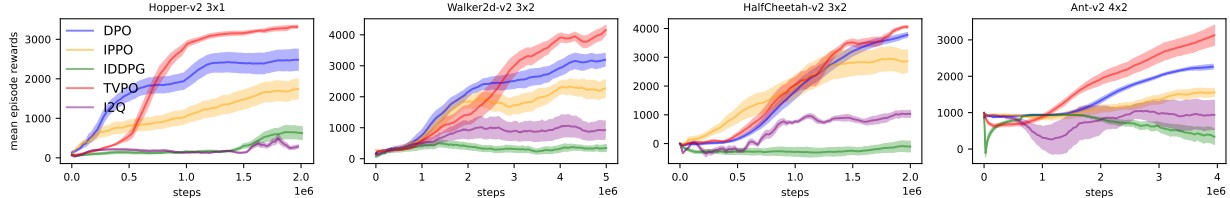

Figure 5: Learning curves of TVPO compared with IDDPG, IPPO I2Q, and DPO in 3-agent Hopper, 3-agent Walker2d, 3-agent HalfCheetah and 4-agent Ant in multi-agent MuJoCo.

## 5.2 Comparing TVPO with DPO

From the discussion in Section 4.2, we have an intuitive idea about the difference between DPO and TVPO that the bound $D_{\text{TV}}$ of TVPO is tighter than $\sqrt{D_{\text{KL}}}$ in DPO. A tighter bound means the iteration will be less influenced by the trivial update. We would like to build a matrix game to show this phenomenon. Fortunately, the matrix game $(a, b, c, d) = (-4, 7, 6, 4)$ satisfies our requirement. The DPO iteration has no closed-form solution and we haven't found any useful properties like Appendix F.4. Thus, we use a numerical method to solve the DPO iteration. First, we keep the initial policy $(p_0, q_0) = (0.55, 0.8)$ for two iterations. The empirical results are included in Figure 2. We can find that the TVPO iteration has a non-trivial update but the DPO iteration only has trivial updates. This result can be evidence for our conclusion about the difference between TVPO and DPO.

Moreover, we study the influence of the initial policies on the DPO iteration. We select three candidate values $C = \{0.2, 0.55, 0.8\}$ for the initial policies. We traverse all the values in $C$ for $(p_0, q_0)$ and conclude the performances of all 9 combinations in Figure 3 and Table 2. We can find all 9 initial policies fall into the trap of the trivial update due to the regularization term $\sqrt{D_{\text{KL}}}$ in DPO. These empirical results can partially exclude the impact of initial policies on the performances of the DPO iteration in this matrix game.

## 5.3 Evaluation of TVPO

We compare TVPO with four baselines: IQL (Tan, 1993), IPPO (de Witt et al., 2020), I2Q (Jiang & Lu, 2022), and DPO (Su & Lu, 2022b). A brief introduction of these baseline algorithms is included in Appendix F.1. In our experiments, all the algorithms use the independent parameter to agree with the fully decentralized setting, and parameter sharing is banned. More details about the experiment settings and hyperparameters are available in Appendix B and D.

**SMAC** is a popular benchmark in cooperative MARL with high-dimensional features and partial observability property. We select five maps in SMAC, 2s3z, 8m, 3s5z, MMM2 and 27m_vs_30m for our experiments. These maps cover all three difficulty levels in SMAC: 2s3z and 8m are easy maps; 3s5z is a hard map; MMM2 and 27m_vs_30m are super-hard maps.

We show the empirical results of these algorithms in Figure 4. In the super-hard maps MMM2 and 27m_vs_30m, all the algorithms can hardly win, so we use episode rewards as the evaluation metric to show the difference more clearly. As illustrated in Figure 4, TVPO has the best performance in all five maps. The performance of DPO and TVPO is similar in the map 8m, and the reason may be that 8m is very easy and both of them can obtain nearly 100% win rates within one million steps. In the other four maps, the differences between TVPO and DPO are more clear.

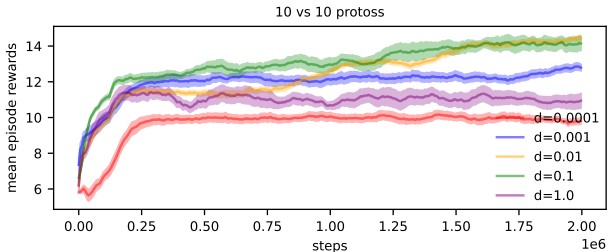

Figure 6: Learning curves of TVPO compared with IQL, IPPO, I2Q, and DPO in 5_vs_5_terran,5_vs_5_protoss,5_vs_5_zerg,10_vs_10_terran,10_vs_10_protoss and 10_vs_10_zerg in SMACv2.

**Multi-Agent MuJoCo** is a robotic locomotion control environment designed for multi-agent scenarios with continuous state and action spaces, based on the single-agent MuJoCo framework (Todorov et al., 2012). In this environment, each agent controls a different part of a robot to perform various tasks. We use independent DDPG (Lillicrap et al., 2016) (IDDPG) to replace IQL for continuous action spaces. As discussed in Section 4.3, we use Hellinger distance to replace total variation distance for continuous action space in TVPO. We select 4 tasks for our experiments: 3-agent Hopper, 3-agent HalfCheetah, 3-agent Walker2d, and 4-agent Ant. In all these tasks, we set agent_obsk=2.

The learning curves of the multi-agent MuJoCo tasks are illustrated in Figure 5. We can find that TVPO substantially outperforms the baselines except in 3-agent HalfCheetah, where DPO obtains similar performance to TVPO. The difference between the performance of the value-based algorithms and the policy-based algorithms is larger in multi-agent MuJoCo compared with SMAC. The reason may be that the continuous action space in fully decentralized learning brings more difficulty in training for the value-based algorithms.

**SMACv2** (Ellis et al., 2023) is a more stochastic and difficult environment based on SMAC, where each agent controls different units and the initial position will also be randomly determined. We select two settings, 5_vs_5 and 10_vs_10, among three races, terran, protoss and zerg, a total of six tasks from SMACv2 in our experiments. The empirical results are illustrated in Figure 6. These tasks are difficult for fully decentralized learning, so we also use the cumulative reward as the metric. We find that TVPO performs better than the four baselines, similar to the results in SMAC.

In all three environments, TVPO obtains the best performance in all the evaluated tasks compared with the four baselines, and the differences between TVPO and the other baselines are obvious in most tasks. The performance of TVPO empirically verifies our discussion about the convergence guarantee of TVPO and the effectiveness of TVPO.

Figure 7: Learning curves of the TVPO with different hyperparameter $d$ in 10_vs_10_protoss in SMAC-v2.

## 5.4 Ablation Study

We select the 10_vs_10_protoss task in SMAC-v2 for the ablation study of the hyperparameters $d$, $\alpha$ and $\beta$. All the learning curves correspond to three random seeds and the shaded area corresponds to 95% confidence interval.

For the ablation study of $d$, we compare the performance of TVPO with $d \in \{0.0001, 0.001, 0.01, 0.1, 1.0\}$. The empirical results are illustrated in Figure 7. Intuitively, $d$ represents the expected distance of $D_{\text{TV}}$ between the old policy and the new policy. If $d$ is small, corresponding to the learning curves $d = 0.0001$ and $d = 0.001$, the step size of the policy update is limited, which may result in relatively low performance. If $d$ is large, corresponding to the learning curves $d = 1.0$, the policy update may exceed the trust region, which is away from the convergence condition and results in oscillating curves. There is a trade-off for $d$. Therefore, the appropriate choices $d = 0.01$ and $d = 0.1$ have the best performance in this task.

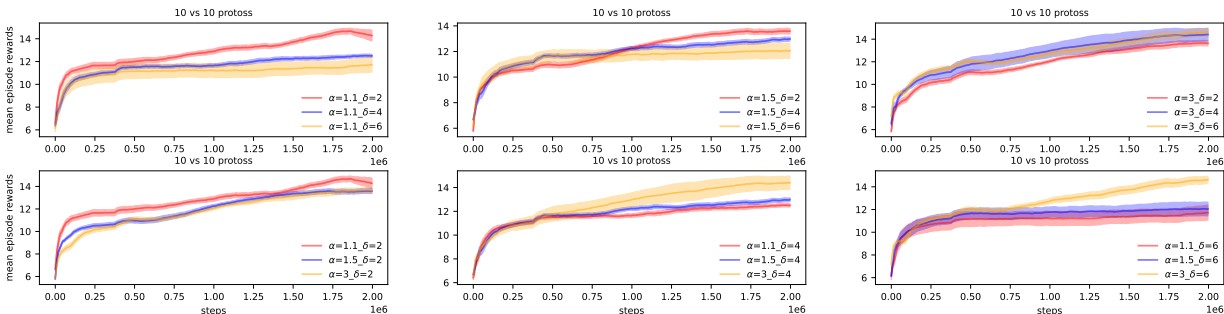

Figure 8: Learning curves of the TVPO with different combinations of hyperparameter $\alpha$ and $\delta$ in 10_vs_10_protoss in SMAC-v2.

For the ablation study of the hyperparameter $\alpha$ and $\delta$, we choose $\alpha \in \{1.1, 1.5, 3\}$ and $\delta \in \{2, 4, 6\}$. The empirical results are shown in Figure 8. In the first line, we control $\alpha$ to be the same in each plot. In the second line, we control $\delta$ to be the same in each plot. Intuitively, $\alpha$ represents the adjustment strength of the coefficient $\beta^i$ and $\delta$ represents the tolerance of the expected distance. A smaller $\delta$ means more frequent adjustments. The empirical results show that $\alpha$ should match $\delta$, *i.e.*, a smaller adjustment strength (a smaller $\alpha$) should correspond to more frequent adjustments (a smaller $\delta$) and vice versa. A good combination of $(\alpha, \delta)$ means a good ability to keep the coefficient $\beta^i$ close to the expected distance $d$. Specifically, among the values of $\alpha$ and $\delta$ we chosen, from the perspective of $\alpha$, $\alpha = 1.1$ and $\alpha = 1.5$ are small values corresponding to the best value $\delta = 2$; from the perspective of $\delta$, $\delta = 4$ and $\delta = 6$ are large values corresponding to the best value $\alpha = 3$.

## 6 Conclusion and Limitations

In this paper, we propose $f$-divergence policy optimization, a general formulation of independent policy optimization in cooperative multi-agent reinforcement learning, and analyze the policy iteration of such a formulation. We discuss the limitation of this formulation, *i.e.*, convergence to only sub-optimal policy, and verify it by the empirical results in a two-player matrix game. Based on $f$-divergence policy optimization, we propose a novel independent learning algorithm, TVPO, and prove its convergence in fully decentralized learning. Empirically, we evaluate TVPO against four baselines in three environments. The empirical results show that TVPO outperforms all the baselines, which verifies the effectiveness of TVPO.

The main limitation of our work is the approximations in the practical algorithms which may not preserve the theoretical properties including the convergence. Additionally, though the learning of decentralized critic is unbiased, it may be troubled with the variance especially in multi-agent settings. Moreover, TVPO still requires on-policy updates which is inconvenient especially in multi-agent settings.

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

# Appendices

## A   Proofs

### A.1   Proof of Lemma 4.2

*Proof.* The Lagrangian function of (4) is as follows:

$$L = \sum_{a_i} \pi^i(a_i|s) Q_i^{\boldsymbol{\pi}_{\text{old}}}(s, a_i) - \omega \sum_{a_i} \pi_{\text{old}}^i(a_i|s) f\left(\frac{\pi^i(a_i|s)}{\pi_{\text{old}}^i(a_i|s)}\right)$$

$$+ \lambda_s \left(\sum_{a_i} \pi^i(a_i|s) - 1\right) + \sum_{a_i} \beta^i(a_i|s)\pi^i(a_i|s),$$

where $\lambda_s$ and $\beta(a_i|s)$ are the Lagrangian multiplier.

Then by the KKT condition we have

$$\frac{\partial L}{\partial \pi^i(a_i|s)} = Q_i^{\boldsymbol{\pi}_{\text{old}}}(s, a_i) - \omega f'\left(\frac{\pi^i(a_i|s)}{\pi_{\text{old}}^i(a_i|s)}\right) + \lambda_s + \beta^i(a_i|s) = 0,$$

so we can resolve $\pi^i(a_i|s)$ as

$$\frac{\pi^i(a_i|s)}{\pi_{\text{old}}^i(a_i|s)} = g\left(\frac{Q_i^{\boldsymbol{\pi}_{\text{old}}}(s, a_i) + \lambda_s + \beta^i(a_i|s)}{\omega}\right) \tag{19}$$

From the complementary slackness we know that $\beta(a_i|s)\pi^i(a_i|s) = 0$, so we can rewrite (19) as

$$\frac{\pi^i(a_i|s)}{\pi_{\text{old}}^i(a_i|s)} = \max\left\{g\left(\frac{Q_i^{\boldsymbol{\pi}_{\text{old}}}(s, a_i) + \lambda_s}{\omega}\right), 0\right\}, \tag{20}$$

$$\pi^i(a_i|s) = \max\left\{\pi_{\text{old}}^i(a_i|s)g\left(\frac{Q_i^{\boldsymbol{\pi}_{\text{old}}}(s, a_i) + \lambda_s}{\omega}\right), 0\right\}. \tag{21}$$

□

### A.2   Proof of Proposition 4.3

*Proof.* To discuss the monotonicity of the policies $p_t$ and $q_t$, let $Q_t^A(0)$ and $Q_t^A(1)$ represent the expected reward Alice will obtain by taking action $u_A^0$ and $u_A^1$ respectively. Simlilarly, we can also define $Q_t^B(0)$ and $Q_t^B(1)$ for Bob.

From the definition, we have $Q_t^A(0) = q_t \cdot a + (1 - q_t) \cdot b = b + (a - b)q_t$. Similarly we can obtain that $Q_t^A(1) = d + (c - d)q_t$, $Q_t^B(0) = c + (a - c)p_t$ and $Q_t^B(1) = d + (b - d)p_t$.

Combining (21) with the condition $g(x) \geq 0$, then we have

$$p_{t+1} = p_t g\left(\frac{(a-b)q_t + b + \lambda_t^A}{\omega}\right),\ 1 - p_{t+1} = (1 - p_t)g\left(\frac{(c-d)q_t + d + \lambda_t^A}{\omega}\right)$$

$$\Rightarrow \frac{1}{p_{t+1}} - 1 = \left(\frac{1}{p_t} - 1\right)\frac{g\left(\frac{(c-d)q_t + d + \lambda_t^A}{\omega}\right)}{g\left(\frac{(a-b)q_t + b + \lambda_t^A}{\omega}\right)}. \tag{22}$$

From (22) we can find that

$$
\begin{aligned}
p_{t+1} \leq p_t \quad &\Leftrightarrow \quad \frac{g\left(\frac{(c-d)q_t+d+\lambda_t^A}{\omega}\right)}{g\left(\frac{(a-b)q_t+b+\lambda_t^A}{\omega}\right)} \geq 1 \\
&\Leftrightarrow \quad (c-d)q_t + d \geq (a-b)q_t + b \\
&\Leftrightarrow \quad (b+c-a-d)q_t \geq b-d \\
&\Leftrightarrow \quad q_t \geq \hat{q}.
\end{aligned}
\tag{23}
$$

The critical step (23) is from the combination of the condition $g(x) \geq 0$ and the property $g(x)$ is non-decreasing.

Similarly we can obtain that $p_t \geq \hat{p} \Rightarrow q_{t+1} \leq q_t$; $p_t \leq \hat{p} \Rightarrow q_{t+1} \geq q_t$; $q_t \geq \hat{q} \Rightarrow p_{t+1} \leq p_t$; and $q_t \leq \hat{q} \Rightarrow p_{t+1} \geq p_t$. $\qquad \square$

### A.3  Proof of Corollary 4.4

*Proof.* From the iteration of $\{p_t\}$ we have

$$
\frac{p_{t+1}}{1-p_{t+1}} = \frac{p_t}{1-p_t} \frac{g\left(\frac{(a-b)q_t+b+\lambda_t^A}{\omega}\right)}{g\left(\frac{(c-d)q_t+d+\lambda_t^A}{\omega}\right)}.
\tag{24}
$$

Let $t \to \infty$ in both side of (24), we know that

$$
\frac{p^*}{1-p^*}\left(\frac{g\left(\frac{(a-b)q^*+b+\lambda_*^A}{\omega}\right)}{g\left(\frac{(c-d)q^*+d+\lambda_*^A}{\omega}\right)} - 1\right) = 0.
\tag{25}
$$

As $q^* > \hat{q}$, we know that $\frac{g\left(\frac{(a-b)q^*+b+\lambda_*^A}{\omega}\right)}{g\left(\frac{(c-d)q^*+d+\lambda_*^A}{\omega}\right)} < 1$. So we can rewrite (25) as $\frac{p^*}{1-p^*} = 0$ and resolve $p^* = 0$.

As for $q^*$, we can follow a similar idea. From the iteration of $\{q_t\}$ we have

$$
\frac{1}{q_{t+1}} - 1 = (\frac{1}{q_t} - 1)\frac{g\left(\frac{(b-d)p_t+d+\lambda_t^B}{\omega}\right)}{g\left(\frac{(a-c)p_t+c+\lambda_t^B}{\omega}\right)}.
\tag{26}
$$

Let $t \to \infty$ in both side of (26), we know that

$$
\frac{1-q^*}{q^*}\left(\frac{g\left(\frac{(b-d)p^*+d+\lambda_*^B}{\omega}\right)}{g\left(\frac{(a-c)p^*+c+\lambda_*^B}{\omega}\right)} - 1\right) = 0.
\tag{27}
$$

As $p^* < \hat{p}$, we know that $\frac{g\left(\frac{(b-d)p^*+d+\lambda_*^B}{\omega}\right)}{g\left(\frac{(a-c)p^*+c+\lambda_*^B}{\omega}\right)} < 1$. Then we can rewrite (27) as $\frac{1-q^*}{q^*} = 0$ and obtain $q^* = 1$. $\quad \square$

### A.4 Proof of Lemma 4.6

*Proof.* For any fixed $i$, consider the following difference

$$\left| \sum_{\boldsymbol{a}} \boldsymbol{\pi}_{\text{new}}(\boldsymbol{a}|s) Q^{\boldsymbol{\pi}}(s, \boldsymbol{a}) - \sum_{a_i} \pi_{\text{new}}^i(a_i|s) \sum_{a_{-i}} \pi_{\text{old}}^{-i}(a_{-i}|s) Q^{\boldsymbol{\pi}}(s, a_i, a_{-i}) \right|$$

$$= \left| \sum_{a_i} \pi_{\text{new}}^i(a_i|s) \sum_{a_{-i}} \left( \pi_{\text{new}}^{-i}(a_{-i}|s) - \pi_{\text{old}}^{-i}(a_{-i}|s) \right) Q^{\boldsymbol{\pi}}(s, a_i, a_{-i}) \right| \tag{28}$$

$$\leq \sum_{a_i} \pi_{\text{new}}^i(a_i|s) \sum_{a_{-i}} \left| \pi_{\text{new}}^{-i}(a_{-i}|s) - \pi_{\text{old}}^{-i}(a_{-i}|s) \right| \left| Q^{\boldsymbol{\pi}}(s, a_i, a_{-i}) \right| \tag{29}$$

$$\leq \frac{M}{2} \sum_{a_i} \pi_{\text{new}}^i(a_i|s) \sum_{a_{-i}} \left| \pi_{\text{new}}^{-i}(a_{-i}|s) - \pi_{\text{old}}^{-i}(a_{-i}|s) \right| \tag{30}$$

$$= \frac{M}{2} \sum_{a_{-i}} \left| \pi_{\text{new}}^{-i}(a_{-i}|s) - \pi_{\text{old}}^{-i}(a_{-i}|s) \right| \tag{31}$$

$$= \frac{M}{2} \sum_{a_{-i}} \left| \sum_{k=1, k\neq i}^{N} \pi_{\text{new}}^{1:k-1}(a_{1:k-1}|s) \pi_{\text{old}}^{k:N}(a_{k:N}|s) - \pi_{\text{new}}^{1:k}(a_{1:k}|s) \pi_{\text{old}}^{k+1\sim N}(a_{k+1:N}|s) \right| \tag{32}$$

$$\leq \frac{M}{2} \sum_{a_{-i}} \sum_{k=1, k\neq i}^{N} \left| \pi_{\text{new}}^{1:k-1}(a_{1:k-1}|s) \pi_{\text{old}}^{k:N}(a_{k:N}|s) - \pi_{\text{new}}^{1:k}(a_{1:k}|s) \pi_{\text{old}}^{k+1\sim N}(a_{k+1:N}|s) \right| \tag{33}$$

$$= \frac{M}{2} \sum_{k=1, k\neq i}^{N} \sum_{a_k} \left| \pi_{\text{new}}^k(a_k|s) - \pi_{\text{old}}^k(a_k|s) \right| \tag{34}$$

$$= M \sum_{k=1, k\neq i}^{N} D_{\text{TV}} \left( \pi_{\text{new}}^k(\cdot|s) \| \pi_{\text{old}}^k(\cdot|s) \right) \tag{35}$$

where $\pi_{\text{new}}^{1:k-1}$ denotes $\pi_{\text{new}}^1 \times \pi_{\text{new}}^2 \times \cdots \pi_{\text{new}}^{k-1}$ and $\pi_{\text{new}}^i$ will be skipped if involved, and $a_{1:k-1}$ has similar meanings as $a_{1:k-1} = a_1 \times a_2 \times \cdots a_{k-1}$. In (29) and (33), we use the triangle inequality of the absolute value. In (30), we use the property $Q^{\boldsymbol{\pi}}(s, \boldsymbol{a}) \leq \frac{r_{\max}}{1-\gamma} = \frac{M}{2}$ from the definition of Q-function. In (32), we insert $N-1$ terms between $\pi_{\text{new}}^{-i}(a_{-i}|s)$ and $\pi_{\text{old}}^{-i}(a_{-i}|s)$ to make sure the adjacent two terms are only different in one individual policy.

By rewriting the conclusion above, for any agent $i$, we have

$$\sum_{\boldsymbol{a}} \boldsymbol{\pi}_{\text{new}}(\boldsymbol{a}|s) Q^{\boldsymbol{\pi}}(s, \boldsymbol{a}) \geq \sum_{a_i} \pi_{\text{new}}^i(a_i|s) \sum_{a_{-i}} \pi_{\text{old}}^{-i}(a_{-i}|s) Q^{\boldsymbol{\pi}}(s, a_i, a_{-i})$$

$$- M \sum_{k=1, k\neq i}^{N} D_{\text{TV}} \left( \pi_{\text{new}}^k(\cdot|s) \| \pi_{\text{old}}^k(\cdot|s) \right). \tag{36}$$

Then, by applying (36) to $i = 1, 2, \cdots, N$ and add all these $N$ inequalities together, we have

$$\sum_{\boldsymbol{a}} \boldsymbol{\pi}_{\text{new}}(\boldsymbol{a}|s) Q^{\boldsymbol{\pi}}(s, \boldsymbol{a}) \geq \frac{1}{N} \sum_{i=1}^{N} \sum_{a_i} \pi_{\text{new}}^i(a_i|s) \sum_{a_{-i}} \pi_{\text{old}}^{-i}(a_{-i}|s) Q^{\boldsymbol{\pi}}(s, a_i, a_{-i})$$

$$- \frac{(N-1)M}{N} \sum_{i=1}^{N} D_{\text{TV}} \left( \pi_{\text{new}}^i(\cdot|s) \| \pi_{\text{old}}^i(\cdot|s) \right).$$

$\square$

## A.5 Proof of Proposition 4.7

*Proof.* By the definition of $V_{\boldsymbol{\rho}}^{\boldsymbol{\pi}_{\text{old}}}$ we have

$$
V_{\boldsymbol{\rho}}^{\boldsymbol{\pi}_{\text{old}}}(s) = \frac{1}{N} \sum_i \sum_{a_i} \pi_{\text{old}}^i(a_i|s) \sum_{a_{-i}} \rho^{-i}(a_{-i}|s) Q_{\boldsymbol{\rho}}^{\boldsymbol{\pi}_{\text{old}}}(s, a_i, a_{-i}) - \omega \sum_i D_f\left(\pi_{\text{old}}^i(\cdot|s) \| \rho^i(\cdot|s)\right)
$$

$$
\leq \frac{1}{N} \sum_i \sum_{a_i} \pi_{\text{new}}^i(a_i|s) \sum_{a_{-i}} \rho^{-i}(a_{-i}|s) Q_{\boldsymbol{\rho}}^{\boldsymbol{\pi}_{\text{old}}}(s, a_i, a_{-i}) - \omega \sum_i D_f\left(\pi_{\text{new}}^i(\cdot|s) \| \rho^i(\cdot|s)\right) \tag{37}
$$

$$
= \frac{1}{N} \sum_i \sum_{a_i} \pi_{\text{new}}^i(a_i|s) \sum_{a_{-i}} \rho^{-i}(a_{-i}|s) \left(r(s, a_i, a_{-i}) + \gamma \mathbb{E}\left[V_{\boldsymbol{\rho}}^{\boldsymbol{\pi}_{\text{old}}}(s')\right]\right)
$$

$$
- \omega \sum_i D_f\left(\pi_{\text{new}}^i(\cdot|s) \| \rho^i(\cdot|s)\right) \tag{38}
$$

$$
\leq \cdots \quad (\text{expand } V_{\boldsymbol{\rho}}^{\boldsymbol{\pi}_{\text{old}}}(s') \text{ and repeat replacing } \pi_{\text{old}}^i \text{ with } \pi_{\text{new}}^i) \tag{39}
$$

$$
\leq V_{\boldsymbol{\rho}}^{\boldsymbol{\pi}_{\text{new}}}(s). \tag{40}
$$

In (37), we use the definition of $\pi_{\text{new}}^i$ in (11). (38) is from the definition of $Q_{\boldsymbol{\rho}}^{\boldsymbol{\pi}_{\text{old}}}(s, a_i, a_{-i})$. In (39), we repeatedly expand $V_{\boldsymbol{\rho}}^{\boldsymbol{\pi}_{\text{old}}}$ according to its definition and replace $\pi_{\text{old}}^i$ with $\pi_{\text{new}}^i$ by the optimality of $\pi_{\text{new}}^i$ like what we have done in (37). After we replace all $\pi_{\text{old}}^i$ with $\pi_{\text{new}}^i$, then we obtain $V_{\boldsymbol{\rho}}^{\boldsymbol{\pi}_{\text{new}}}(s)$ according to the definition of $V_{\boldsymbol{\rho}}^{\boldsymbol{\pi}_{\text{new}}}(s)$ in (40).

With the result $V_{\boldsymbol{\rho}}^{\boldsymbol{\pi}_{\text{old}}}(s) \leq V_{\boldsymbol{\rho}}^{\boldsymbol{\pi}_{\text{new}}}(s)$, we know $Q_{\boldsymbol{\rho}}^{\boldsymbol{\pi}_{\text{old}}}(s, \boldsymbol{a}) = r(s, \boldsymbol{a}) + \gamma \mathbb{E}[V_{\boldsymbol{\rho}}^{\boldsymbol{\pi}_{\text{old}}}(s')] \leq r(s, \boldsymbol{a}) + \gamma \mathbb{E}[V_{\boldsymbol{\rho}}^{\boldsymbol{\pi}_{\text{new}}}(s')] = Q_{\boldsymbol{\rho}}^{\boldsymbol{\pi}_{\text{new}}}(s, \boldsymbol{a})$. $\square$

## A.6 Proof of Theorem 4.8

*Proof.* From the Proposition 4.7, we know $V_{\boldsymbol{\pi}_t}^{\boldsymbol{\pi}_{t+1}}(s) \geq V^{\boldsymbol{\pi}_t}(s)$. Thus, we just need to prove $V^{\boldsymbol{\pi}_t}(s) \geq V_{\boldsymbol{\pi}_{t-1}}^{\boldsymbol{\pi}_t}(s)$. From the definition of $V^{\boldsymbol{\pi}_t}(s)$ we have

$$
V^{\boldsymbol{\pi}_t}(s) = \sum_{\boldsymbol{a}} \boldsymbol{\pi}_t(\boldsymbol{a}|s) Q^{\boldsymbol{\pi}_t}(s, \boldsymbol{a})
$$

$$
\geq \frac{1}{N} \sum_{i=1}^N \sum_{a_i} \pi_t^i(a_i|s) \sum_{a_{-i}} \pi_{t-1}^{-i}(a_{-i}|s) Q^{\boldsymbol{\pi}_t}(s, a_i, a_{-i})
$$

$$
- \omega \sum_{i=1}^N D_{\text{TV}}\left(\pi_t^i(\cdot|s) \| \pi_{t-1}^i(\cdot|s)\right) \tag{41}
$$

$$
= \frac{1}{N} \sum_{i=1}^N \sum_{a_i} \pi_t^i(a_i|s) \sum_{a_{-i}} \pi_{t-1}^{-i}(a_{-i}|s) \left(r(s, a_i, a_{-i}) + \gamma \mathbb{E}[V^{\boldsymbol{\pi}_t}(s')]\right)
$$

$$
- \omega \sum_{i=1}^N D_{\text{TV}}\left(\pi_t^i(\cdot|s) \| \pi_{t-1}^i(\cdot|s)\right) \tag{42}
$$

$$
\geq \cdots \quad (\text{expand } V^{\boldsymbol{\pi}_t}(s') \text{ and repeat replacing } \pi_t^{-i} \text{ with } \pi_{t-1}^{-i}) \tag{43}
$$

$$
\geq V_{\boldsymbol{\pi}_{t-1}}^{\boldsymbol{\pi}_t}(s). \tag{44}
$$

(41) is from Lemma 4.6, and (42) is from the definition of $Q^{\boldsymbol{\pi}_t}(s, a_i, a_{-i})$. In (43), we repeatedly expand $V^{\boldsymbol{\pi}_t}$ and replace the $\pi_t^{-i}$ with $\pi_{t-1}^{-i}$ by Lemma 4.6 like what we have done in (41). After we replace all $\pi_t^{-i}$ with $\pi_{t-1}^{-i}$, then we obtain $V_{\boldsymbol{\pi}_{t-1}}^{\boldsymbol{\pi}_t}(s)$ in (44) according to the definition of $V_{\boldsymbol{\pi}_{t-1}}^{\boldsymbol{\pi}_t}(s)$.

From the inequalities $V_{\boldsymbol{\pi}_t}^{\boldsymbol{\pi}_{t+1}}(s) \geq V^{\boldsymbol{\pi}_t}(s) \geq V_{\boldsymbol{\pi}_{t-1}}^{\boldsymbol{\pi}_t}(s) \geq V^{\boldsymbol{\pi}_{t-1}}(s)$, we know that the sequence $\{V^{\boldsymbol{\pi}_t}\}$ improves monotonically. Combining with the condition that the sequence $\{V^{\boldsymbol{\pi}_t}\}$ is bounded, we know that $\{V^{\boldsymbol{\pi}_t}\}$ will converge to $V^*$. According to the definition, the sequence $\{Q^{\boldsymbol{\pi}_t}\}$ and $\{\boldsymbol{\pi}_t\}$ will also converge to $Q^*$ and $\boldsymbol{\pi}_*$

respectively, where $\boldsymbol{\pi}_*$ satisfies the following fixed-point equation:

$$\pi_*^i = \arg\max_{\pi^i} \sum_{a_i} \pi^i(a_i|s) \sum_{a_{-i}} \pi_*^{-i}(a_{-i}|s) Q^*(s, a_i, a_{-i}) - \omega D_{\mathrm{TV}}\left(\pi^i(\cdot|s)\|\pi_*^i(\cdot|s)\right).$$

□

### A.7 Proof of $D_{\mathrm{TV}}(p\|q) \leq D_{\mathrm{H}}(p\|q)$

*Proof.*

$$
\begin{aligned}
D_{\mathrm{TV}}^2(p\|q) &= \frac{1}{4}\left(\sum_i |p_i - q_i|\right)^2 = \frac{1}{4}\left(\sum_i |\sqrt{p_i} - \sqrt{q_i}|\,|\sqrt{p_i} + \sqrt{q_i}|\right)^2 \\
&\leq \frac{1}{4}\left(\sum_i |\sqrt{p_i} - \sqrt{q_i}|^2\right)\left(\sum_i |\sqrt{p_i} + \sqrt{q_i}|^2\right) \quad \text{(Cauchy–Schwarz inequality)} \\
&= \frac{1}{4} D_{\mathrm{H}}^2(p\|q)\left(2 + 2\sum_i \sqrt{p_i q_i}\right) \\
&\leq D_{\mathrm{H}}^2(p\|q).
\end{aligned}
$$

□

## B Experimental Settings

### B.1 MPE

The three tasks are based on the original Multi-Agent Particle Environment (MPE) (Lowe et al., 2017) (MIT license) and were initially used in Agarwal et al. (2020) (MIT license). The objectives of these tasks are:

- **Simple Spread:** $N$ agents must occupy the locations of $N$ landmarks.

- **Line Control:** $N$ agents must line up between two landmarks.

- **Circle Control:** $N$ agents must form a circle around a landmark.

The reward in these tasks is the distance between all the agents and their target locations. We select these tasks to maintain consistency with DPO (Su & Lu, 2022b) but set the number of agents $N = 10$ for these three tasks in our experiment.

### B.2 Multi-Agent MuJoCo

Multi-agent MuJoCo (Peng et al., 2021) (Apache-2.0 license) is a robotic locomotion task featuring continuous action space for multi-agent settings. The robot is divided into several parts, each containing multiple joints. Agents in this environment control different parts of the robot. The type of robot and the assignment of joints determine the task. For example, the task "HalfCheetah-3×2" means dividing the robot "HalfCheetah" into three parts, with each part containing two joints. Details of our experiment settings in multi-agent MuJoCo are listed in Table 3. The configuration specifies the number of agents and the joints assigned to each agent. "Agent obsk" defines the number of nearest agents an agent can observe.

### B.3 StarCraft2

SMAC (Samvelyan et al., 2019) (MIT license) is a widely used environment for multi-agent reinforcement learning (MARL). In SMAC, agents receive rewards when they attack or kill an enemy unit. The rewards for an episode are normalized to a maximum of 20, regardless of the number of agents, to ensure consistency

Table 3: The task settings of multi-agent MuJoCo

| task | configuration | agent obsk |
|---|---|---|
| HalfCheetah | $3\times2$ | 2 |
| Hopper | $3\times1$ | 2 |
| Walker2d | $3\times2$ | 2 |
| Ant | $4\times2$ | 2 |

across tasks. An episode is considered won if the agents kill all enemy units. The observation space for agents depends on the number of units involved in the task. Typically, the observation is a vector with over 100 dimensions, containing information about all units. Information about units outside an agent's field of view is represented as zero in the observation vector. More details on SMAC can be found in the original paper (Samvelyan et al., 2019). SMACv2 (Ellis et al., 2023) (MIT license) is an advanced version of SMAC. Unlike SMAC, SMACv2 allows agents to control different types of units in different episodes, where the unit types are determined by a distribution and a type list. Moreover, the initial positions of agents are randomly selected in different episodes. With these properties, SMACv2 is more stochastic and difficult than SMAC. We keep the configuration the same as the original paper (Ellis et al., 2023) among the selected tasks.

## C  Algorithm

---

**Algorithm 1.** The practical algorithm of TVPO

---

1: **for** episode $= 1$ to $M$ **do**
2:     **for** $t = 1$ to max_episode_length **do**
3:         select action $a_i \sim \pi^i(\cdot|s)$
4:         execute $a_i$ and observe reward $r$ and next state $s'$
5:         collect $\langle s, a_i, r, s' \rangle$
6:     **end for**
7:     Update the critic according to (17)
8:     Update the policy according to (15) or (18)
9:     Update $\beta^i$ according to (16).
10: **end for**

---

## D  Training Details

Our code of IPPO is based on the open-source code[1] of MAPPO (Yu et al., 2021) (MIT license). The original IPPO and MAPPO is actually implemented as a CTDE method with parameter sharing and centralized critics. We modify the code for individual parameters and ban the tricks used by MAPPO for SMAC. The network architectures and base hyperparameters of TVPO, DPO and IPPO are the same for all the tasks in all the environments. We use 3-layer MLPs for the actor and the critic and use ReLU as non-linearities. The number of the hidden units of the MLP is 128. We train all the networks with an Adam optimizer. The learning rates of the actor and critic are both 5e-4. The number of epochs for every batch of samples is 15 which is the recommended value in Yu et al. (2021). For IPPO, the clip parameter is 0.2 which is the same as Schulman et al. (2017). For DPO, the hyperparameter is set as the original paper (Su & Lu, 2022b) recommends. Our code of IQL is based on the open-source code[2] PyMARL (Apache-2.0 license) and we modify the code for individual parameters. The default architecture in PyMARL is RNN so we just follow it and the number of the hidden units is 128. The learning rate of IQL is also 5e-4. The architectures of the actor and critic of IDDPG are 3-layer MLPs. The learning rates of the actor and critic are both

---

[1]https://github.com/marlbenchmark/on-policy
[2]https://github.com/oxwhirl/pymarl

5e-4. Our code of I2Q is from the open source code[3] of the original paper (Jiang & Lu, 2022). We keep the hyperparameter of I2Q the same as the default value of the open-source code in our experiments.

Table 4: Hyperparameters for all the experiments

| hyperparameter | value |
|---|---|
| MLP layers | 3 |
| hidden size | 128 |
| non-linear | ReLU |
| optimizer | Adam |
| actor_lr | 5e-4 |
| critic_lr | 5e-4 |
| numbers of epochs | 15 |
| initial $\beta^i$ | 0.01 |
| $\delta$ | 1.5 |
| $\omega$ | 2 |
| $d$ | 0.001 |
| clip parameter for IPPO | 0.2 |

The version of the game StarCraft2 in SMAC is 4.10 for our experiments in all the SMAC tasks. We set the episode length of all the multi-agent MuJoCo tasks as 1000 in all of our multi-agent MuJoCo experiments. We perform the whole experiment with a total of four NVIDIA A100 GPUs. We have summarized the hyperparameters in Table 4.

## E  Additional Empirical Results

Figure 9 illustrates the learning curve of the policy $p$ and $q$ in the matrix game of KL-iteration, TV-iteration, $\chi^2$-iteration, and H-iteration over four different sets of initialization. We can observe the policies of all four kinds of iterations converge.

MPE is a popular environment in cooperative MARL. MPE is a 2D environment and the objects are either agents or landmarks. Landmark is a part of the environment, while agents can move in any direction. With the relation between agents and landmarks, we can design different tasks. We use the discrete action space version of MPE and the agents can accelerate or decelerate in the direction of the x-axis or y-axis. We choose MPE for its partial observability.

The empirical results in MPE are illustrated in Figure 10. We find that TVPO obtains the best performance in all three tasks. In this environment, the policy-based algorithms, TVPO, DPO, and IPPO, outperform the value-based algorithms, IQL and I2Q. I2Q has a better performance than IQL in all three tasks.

For the comparison with the baseline IPG (Leonardos et al., 2021) and INPG (Fox et al., 2022), we select three 10_vs_10 SMAC-v2 tasks. The empirical results are illustrated Figure 11. We can find that IPG's performance is not stationary and may drop with the progress of training compared with other policy based algorithms. We think the main reason is that IPG lack the constraints about the stepsize of policy iteration. We use the adaptive coefficient for INPG, and its performance is similar to DPO, which is reasonable as their policy objectives are similar except for a square root term.

We also compare the influence of the hyperparameters on IPPO's performance. We choose clip parameters with values $0.1, 0.2, 0.3$ for ablation study and select the 10_vs_10 protoss task for experiments. The empirical results are ilustrated in Figure 12. We can see that the impact of this hyperparameter is not significant.

---

[3]https://github.com/jiechuanjiang/I2Q

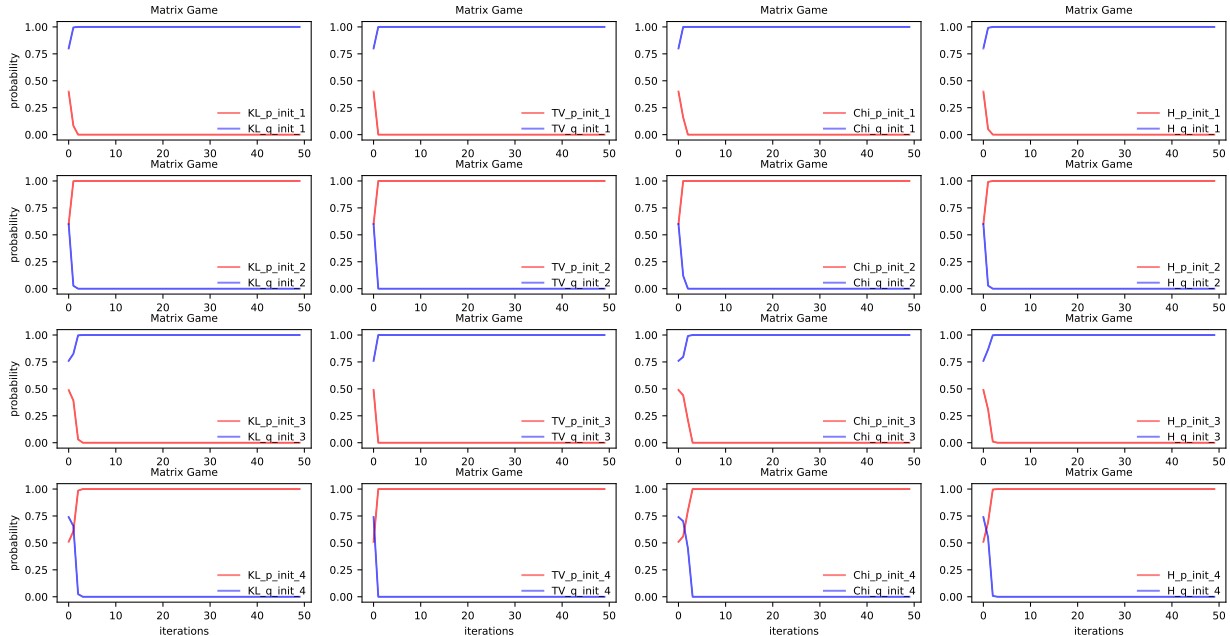

Figure 9: Learning curves of the policy $p$ and $q$ in the matrix game of KL-iteration, TV-iteration, $\chi^2$-iteration, and H-iteration over four different sets of initialization. Each row corresponds to one set of initialization and each column corresponds to one type of iteration.

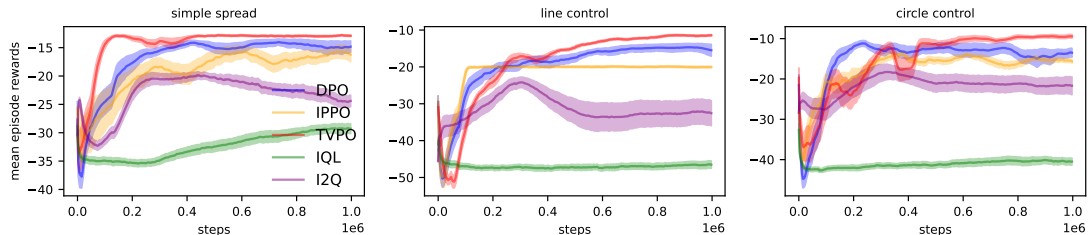

Figure 10: Learning curves of TVPO compared with IQL, IPPO, I2Q, and DPO in 10-agent simple spread, 10-agent line control, and 10-agent circle control in MPE.

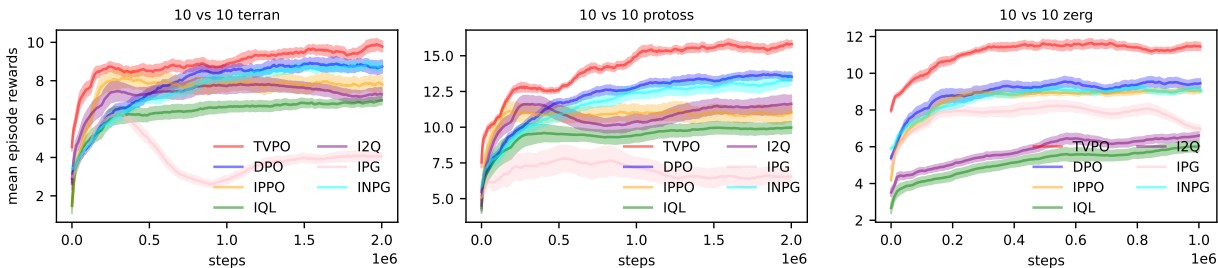

Figure 11: Learning curves of the TVPO and other baselines including IPG and INPG in the three 10_vs_10 SMAC-v2 tasks.

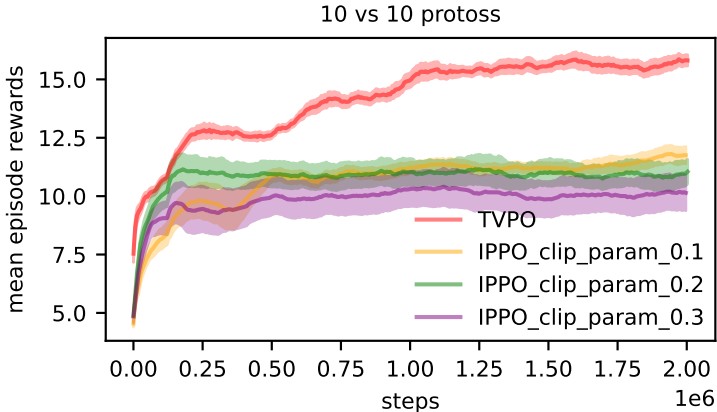

Figure 12: Learning curves of the TVPO and IPPO with different clip parameters in the 10__vs__10 protoss.

## F  Discussion

### F.1  A Brief introduction of baseline algorithms

We select these four baseline algorithms as representatives of fully decentralized algorithms. IQL (Tan, 1993) is a basic value-based algorithm for decentralized learning. IPPO is a basic policy-based algorithm for decentralized learning. Both IQL and IPPO (de Witt et al., 2020) do not have convergence guarantees, to the best of our knowledge. DPO (Su & Lu, 2022b) and I2Q (Jiang & Lu, 2022) are the recent policy-based algorithm and value-based algorithm respectively, and both of them have been proved to have convergence guarantee.

IQL, IDDPG, and IPPO are relatively simple to understand, where each agent updates its policy through an independent Q-learning, DDPG, or PPO. These algorithms simply extend the single-agent RL algorithms into the MARL setting. They are heuristic algorithms without convergence guarantees in fully decentralized MARL.

The idea of DPO is to find a lower bound of the joint policy improvement objective as a surrogate which can also be optimized in a decentralized way for each agent. The formulation of DPO is as follows:

$$\pi_{t+1}^i == \arg\max_{\pi^i} \sum_{a_i} \pi^i(a_i|s) Q_i^{\boldsymbol{\pi}_t}(s, a_i) - \hat{M} \cdot \sqrt{D_{\mathrm{KL}}\left(\pi^i(\cdot|s)\|\pi_t^i(\cdot|s)\right)} - C \cdot D_{\mathrm{KL}}\left(\pi^i(\cdot|s)\|\pi_t^i(\cdot|s)\right).$$

DPO has been proven to improve monotonically and converge in fully decentralized MARL.

I2Q uses Q-learning from the perspective of QSS-value $Q_i(s, s')$. The QSS-value is updated with the following operator:

$\Gamma Q_i(s, s') = r + \gamma \max_{s'' \in \mathcal{N}(s')} Q_i(s', s'')$,

where $\mathcal{N}(s')$ is the neighbor set of state $s'$. In the deterministic environment and with some assumption about the transition probability, $Q_i(s, s')$ will converge to the same Q-function for each agent $i$, so the joint policy of agents will also converge in fully decentralized MARL.

### F.2  Unary Formulation

Before proposing the $f$-divergence formulation, we have studied another formulation. This formulation follows the idea of entropy regularization and the extra term is only related to the policy $\pi^i$ instead of the divergence between $\pi^i$ and $\pi_{\mathrm{old}}^i$. We refer to this approach as the unary formulation. Though we discovered that the unary formulation has more significant drawbacks, the properties of the unary formulation inspire us in

the proof of TVPO. So we would like to provide the properties and some empirical results of the unary formulation here for discussion.

The unary formulation is

$$\pi_{\text{new}}^i = \arg\max_{\pi^i} \sum_{a_i} \pi^i(a_i|s) Q_i^{\boldsymbol{\pi}_{\text{old}}}(s, a_i) + \omega \sum_{a_i} \pi^i(a_i|s) \phi\left(\pi^i(a_i|s)\right). \tag{45}$$

This formulation (45) follows the idea of Yang et al. (2019) which discusses the regularization algorithm in single-agent RL. From the perspective of regularization, the update rule (45) can be seen as optimizing the regularized objective $J_\phi^i(\boldsymbol{\pi}) = \mathbb{E}\left[\sum_t \gamma^t \left(r_i(s, a_i) + \omega\phi\left(\pi^i(a_i|s)\right)\right)\right]$, where $r_i(s, a_i) = \mathbb{E}_{\pi^{-i}}\left[r(s, a_i, a_{-i})\right]$. The choice of $\phi$ is flexible, *e.g.*, $\phi(x) = -\log x$ corresponds to entropy regularization and independent SAC (Haarnoja et al., 2018); $\phi(x) = 0$ means (45) degenerates to independent Q-learning (Tan, 1993); Moreover, there are many other options for $\phi$ corresponding to different regularization (Yang et al., 2019). So we take (45) as the general unary formulation of independent learning, where the 'unary' means the additional terms $\sum_{a_i} \pi^i(a_i|s)\phi\left(\pi^i(a_i|s)\right)$ is only about one policy $\pi^i$.

For further discussion of (45), we can utilize the conclusion in Yang et al. (2019) as the following lemma.

**Lemma F.1.** *If $\phi(x)$ in $(0, 1]$ and satisfies the following conditions: (1) $\phi(x)$ is non-increasing; (2) $\phi(1) = 0$; (3) $\phi(x)$ is differentiable; (4) $f_\phi(x) = x\phi(x)$ is strictly concave, then we have that $g_\phi(x) = (f'_\phi)^{-1}(x)$ exists and $g_\phi(x)$ is decreasing. Moreover, the solution to the optimization objective (45) can be described with $g_\phi(x)$ as follows:*

$$\pi_{\text{new}}^i(a_i|s) = \max\{g_\phi\left(\frac{\lambda_s - Q_i^{\boldsymbol{\pi}_{\text{old}}}(s, a_i)}{\omega}\right), 0\}, \tag{46}$$

*where $\lambda_s$ satisfies $\sum_{a_i} \max\{g_\phi\left(\frac{\lambda_s - Q_i^{\boldsymbol{\pi}_{\text{old}}}(s, a_i)}{\omega}\right), 0\} = 0$.*

Though it seems that $\phi(x)$ needs to satisfy four conditions, actually $\phi(x) = -\log x$ for Shannon entropy and $\phi(x) = \frac{k}{q-1}(1 - x^{q-1})$ for Tsallis entropy are still qualified.

However, unlike the single-agent setting, the update rule in Lemma F.1 may result in the convergence to sub-optimal policy or even oscillations in policy in fully decentralized MARL.

We further discuss (45) in the two-player matrix game and have the following proposition.

**Proposition F.2.** *Suppose that $g_\phi(x) \geq 0$ and $g_\phi(x)$ is continuously differentiable. If the payoff matrix of the two-player matrix game satisfies $b + c < a + d$, and two agents Alice and Bob update their policies with policy iteration as*

$$\pi_{t+1}^i = \arg\max_{\pi^i} \sum_{a_i} \pi^i(a_i|s) Q_i^{\boldsymbol{\pi}_t}(s, a_i) + \omega \sum_{a_i} \pi^i(a_i|s) \phi\left(\pi^i(a_i|s)\right), \tag{47}$$

*then we have (1) $p_t > p_{t-1} \Rightarrow q_{t+1} > q_t$; (2) $p_t < p_{t-1} \Rightarrow q_{t+1} < q_t$; (3) $q_t > q_{t-1} \Rightarrow p_{t+1} > p_t$; (4) $q_t < q_{t-1} \Rightarrow p_{t+1} < p_t$.*

*Proof.* To discuss the monotonicity of the policies $p_t$ and $q_t$, we need the solution in Lemma F.1. Before applying the update rule (46), we need to calculate the decentralized critic given $p_t$ and $q_t$. Let $Q_t^A(0)$ and $Q_t^A(1)$ represent the expected reward Alice will obtain by taking action $u_A^0$ and $u_A^1$ respectively. We can also define $Q_t^B(0)$ and $Q_t^B(1)$ for Bob.

From the definition, we have $Q_t^A(0) = q_t \cdot a + (1 - q_t) \cdot b = b + (a - b)q_t$. Similarly we could obtain that $Q_t^A(1) = d + (c - d)q_t$, $Q_t^B(0) = c + (a - c)p_t$ and $Q_t^B(1) = d + (b - d)p_t$.

With (46) and the condition $g_\phi(x) \geq 0$, we have

$$p_{t+1} = g_\phi \left( \frac{\lambda_t^A - Q_t^A(0)}{\omega} \right) = g_\phi \left( \frac{(b-a)q_t + \lambda_t^A - b}{\omega} \right), \ 1 - p_{t+1} = g_\phi \left( \frac{(d-c)q_t + \lambda_t^A - d}{\omega} \right)$$

$$g_\phi \left( \frac{(b-a)q_t + \lambda_t^A - b}{\omega} \right) + g_\phi \left( \frac{(d-c)q_t + \lambda_t^A - d}{\omega} \right) = 1$$

$$q_{t+1} = g_\phi \left( \frac{(c-a)p_t + \lambda_t^B - c}{\omega} \right), \ 1 - q_{t+1} = g_\phi \left( \frac{(d-b)p_t + \lambda_t^B - d}{\omega} \right)$$

$$g_\phi \left( \frac{(c-a)p_t + \lambda_t^B - c}{\omega} \right) + g_\phi \left( \frac{(d-b)p_t + \lambda_t^B - d}{\omega} \right) = 1.$$

We can rewrite these equations with some simplifications as follows,

$$m_A(x) \triangleq \frac{(b-a)x + \lambda_A(x) - b}{\omega}, \ n_A(x) \triangleq \frac{(d-c)x + \lambda_A(x) - d}{\omega}, \ h_A(x) = g_\phi(m_A(x))$$

$$\text{where } \lambda_A(x) \text{ satisfies } g_\phi(m_A(x)) + g_\phi(n_A(x)) = 1 \tag{48}$$

$$m_B(x) \triangleq \frac{(c-a)p_t + \lambda_B(x) - c}{\omega}, \ n_B(x) \triangleq \frac{(d-b)p_t + \lambda_B(x) - d}{\omega}, \ h_B(x) = g_\phi(m_B(x))$$

$$\text{where } \lambda_B(x) \text{ satisfies } g_\phi(m_B(x)) + g_\phi(n_B(x)) = 1.$$

With these definitions, we know that $p_{t+1} = h_A(q_t)$, $q_{t+1} = h_B(p_t)$ and the monotonicity of $p_t$ and $q_t$ is determined by the property of function $h_A(x)$ and $h_B(x)$. By applying the chain rule to (48), we have:

$$\frac{1}{\omega} g'_\phi(m_A(x))(b - a + \lambda'_A(x)) + \frac{1}{\omega} g'_\phi(n_A(x))(d - c + \lambda'_A(x)) = 0$$

$$\Rightarrow \lambda'_A(x) = -\frac{(b-a)g'_\phi(m_A(x)) + (d-c)g'_\phi(n_A(x))}{g'_\phi(m_A(x)) + g'_\phi(n_A(x))}. \tag{49}$$

Then we have:

$$h'_A(x) = \frac{1}{\omega} g'_\phi(m_A(x))(b - a + \lambda'_A(x)) \quad \text{(Apply chain rule)} \tag{50}$$

$$= \frac{1}{\omega}(b + c - a - d) \frac{g'_\phi(n_A(x))g'_\phi(m_A(x))}{g'_\phi(m_A(x)) + g'_\phi(n_A(x))} \quad \text{(Substitute (49) for } \lambda'_A(x) \text{)}. \tag{51}$$

Let $M = b + c - a - d$ and $M' = \frac{M}{\omega}$, then $h'_A(x) = M' \frac{g'_\phi(n_A(x))g'_\phi(m_A(x))}{g'_\phi(m_A(x)) + g'_\phi(n_A(x))}$. From the condition and Lemma F.1 we know that $M' < 0$ and $g_\phi(x)$ is decreasing which means $g'_\phi(x) < 0$. Combining these conditions together, we know $h'_A(x) > 0$ and $h_A(x)$ is increasing which means that $p_{t+1} = h_A(q_t)$ is increasing over $q_t$, which means that $q_t > q_{t-1} \Rightarrow p_{t+1} > p_t$ and $q_t > q_{t-1} \Rightarrow p_{t+1} > p_t$.

Similarly, we can obtain that $h'_B(x) = M' \frac{g'_\phi(n_B(x))g'_\phi(m_B(x))}{g'_\phi(m_B(x)) + g'_\phi(n_B(x))} > 0$ which could lead to the result that $p_t > p_{t-1} \Rightarrow q_{t+1} > q_t$ and $p_t < p_{t-1} \Rightarrow q_{t+1} < q_t$. □

Proposition F.2 actually tells us $p_{t+1} = h_A(q_t)$ is increasing over $q_t$ and $q_{t+1} = h_B(p_t)$ is increasing over $p_t$ when $M = b + c - a - d < 0$. Intuitively, we can find two typical cases for policy iterations with Proposition F.2. In the first case, if in a certain iteration $t$ the conditions $p_t > p_{t-1}$ and $q_t > q_{t-1}$ are satisfied, then we know that $p_{t'+1} > p_{t'} \ q_{t'+1} > q_{t'} \ \forall t' \geq t$. As the sequences $\{p_t\}$ and $\{q_t\}$ are both bounded in the interval $[0,1]$, we know that $\{p_t\}$ and $\{q_t\}$ will converge to $p^*$ and $q^*$. The property of $p^*$ and $q^*$ is determined by $l_A(x) \triangleq h_B(h_A(x))$ and $l_B(x) \triangleq h_A(h_B(x))$ respectively as $p_{t+2} = h_B(h_A(p_t))$ and $q_{t+2} = h_A(h_B(q_t))$ and we have the following corollary.

**Corollary F.3.** $|l'_A(x)| \leq M'^2 U_\phi^2$, $|l'_B(x)| \leq M'^2 U_\phi^2$, where $U_\phi$ is a constant determined by $\phi(x)$.

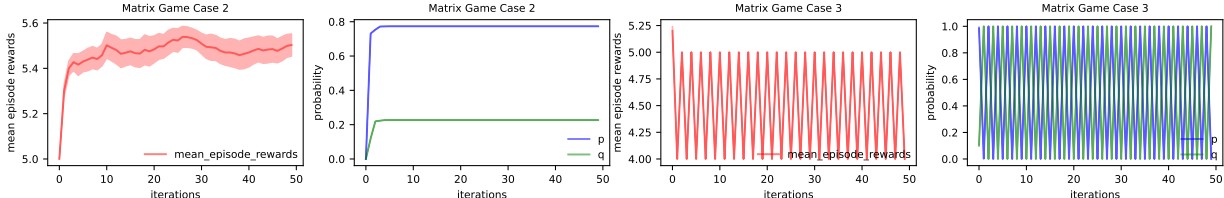

Figure 13: Learning curves of the unary formulation in two matrix game cases, where x-axis is iteration steps. The first and second figures show the performance and the policies $p$ and $q$ in the matrix game case 2 respectively. The third and fourth figures show the performance and the policies $p$ and $q$ in the matrix game case 3 respectively.

*Proof.* As $g'_\phi(x)$ is continuous, let $U^1_A \triangleq \max_{x\in[0,1]} |g'_\phi(m_A(x))|$, $U^2_A \triangleq \max_{x\in[0,1]} |g'_\phi(n_A(x))|$, $U^1_B \triangleq \max_{x\in[0,1]} |g'_\phi(m_B(x))|$ and $U^2_B \triangleq \max_{x\in[0,1]} |g'_\phi(n_B(x))|$. Moreover, let $U_\phi = \max\{U^1_A, U^2_A, U^1_B, U^2_B\}$, then apply the chain rule to $l'_A(x)$ and we have

$$
\begin{aligned}
|l'_A(x)| &= |h'_B(h_A(x))h'_A(x)| \\
&= M'^2 \frac{|g'_\phi(n_B(h_A(x)))||g'_\phi(m_B(h_A(x)))|}{|g'_\phi(m_B(h_A(x)))| + |g'_\phi(n_B(h_A(x)))|} \frac{|g'_\phi(n_A(x))||g'_\phi(m_A(x))|}{|g'_\phi(m_A(x))| + |g'_\phi(n_A(x))|} \qquad (52) \\
&= M'^2 \frac{|g'_\phi(n_B(y))||g'_\phi(m_B(y))|}{|g'_\phi(m_B(y))| + |g'_\phi(n_B(y))|} \frac{|g'_\phi(n_A(x))||g'_\phi(m_A(x))|}{|g'_\phi(m_A(x))| + |g'_\phi(n_A(x))|} \quad (\text{Let } y = h_A(x) \in [0,1]) \\
&\le M'^2 \frac{|g'_\phi(m_B(y))| + |g'_\phi(n_B(y))|}{2} \frac{|g'_\phi(m_A(x))| + |g'_\phi(n_A(x))|}{2} \qquad (53) \\
&\le M'^2 U_\phi^2 \qquad (54)
\end{aligned}
$$

where (52) is from Proposition F.2, (53) is from the AM-GM inequality $ab \le \frac{(a+b)^2}{2}$, and (54) is from the definition of $U_\phi$. Similarly, we can obtain $|l'_B(x)| \le M'^2 U_\phi^2$. $\qquad \square$

Combining Corollary F.3 and Banach fixed-point theorem, we can find that as $U_\phi$ is a constant, if $|M'| < \frac{1}{U_\phi}$, then we can find a constant $L$ such that $|l'_A(x)| \le M'^2 U_\phi^2 \le L < 1$, which means that the iteration $p_{t+1} = l_A(p_t)$ is a contraction and $p^*$ is the unique fixed-point of $l_A$. This conclusion can be seen as that a smaller $|M'|$ corresponds to a larger probability of convergence. In this convergence case, the converged policies $p^*$ and $q^*$ are usually not the optimal policy as the optimal policy is deterministic, which can be seen in our empirical results.

In the second case, which may be more general, in iteration $t$, $(p_t - p_{t-1})(q_t - q_{t-1}) < 0$, which means $p_t > p_{t-1}$ and $q_t < q_{t-1}$ or $p_t < p_{t-1}$ and $q_t > q_{t-1}$. Without loss of generality, we assume $p_t > p_{t-1}$ and $q_t < q_{t-1}$, then we know $p_{t+1} < p_t$ and $q_{t+1} < q_t$ from Proposition F.2. By induction we can find that for any $t' \ge t$, the sequence $\{p_{t'}\}$ and $\{q_{t'}\}$ will increase and decrease alternatively, which means that the policies may not converge but oscillate. We will show this in our experiments. As the unary formulation may result in policy oscillation, we would like to find other formulations for fully decentralized MARL.

### F.3 Verification for Unary Formulation

In this section, we choose $\phi(x) = -\log x$ corresponding to the entropy regularization as the representation for the unary formulation. We build two cases to show the convergence to the sub-optimal policy and the policy oscillation. We choose $a = 5, b = 6, c = 3, d = 5$ as case 2 and $a = 7, b = 5, c = 4, d = 6$ as case 3. Both two cases satisfy the condition $b + c < a + d$ as discussed above. We keep $\omega = 0.1$ for all the experiments on these two matrix games. The empirical results are illustrated in Figure 13. We can find the policies $p$ and $q$ improve monotonically to the convergence $(p^*, q^*) \approx (0.773, 0.227)$ in case 2, which is a sub-optimal joint policy. However, in case 3, the policies $p$ and $q$ oscillate between 0 and 1 and do not converge. These results verify our discussion about the limitation of the unary formulation.

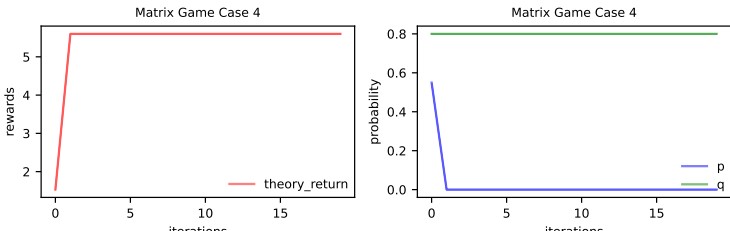

Figure 14: Learning curves of the iteration (13) in the matrix game $(a, b, c, d) = (-4, 7, 6, 4)$, where x-axis is iteration steps. The first and second figures show the expectation $J(\boldsymbol{\pi}_t)$ and the policies $p$ and $q$ in the matrix game case 4 respectively, where $J(\boldsymbol{\pi}_t)$ is calculated by the joint policy $\boldsymbol{\pi}_t = (p_t, q_t)$ and the payoff matrix.

### F.4 Non-Trivial Solution to Iteration (13)

In this section, we will build a two-player matrix game like Table 1 to show the non-trivial solution to iteration (13). In general, there is no closed-form solution to iteration (13). However, for the matrix game case, we can show some properties of iteration (13). With the same definitions as previous discussions, we can rewrite (13) in the matrix game as follows:

$$p_{t+1} = \arg\max_{p \in [0,1]} p Q_t^A(0) + (1-p) Q_t^A(1) - \omega |p - p_t|. \tag{55}$$

Let $f(p) = p Q_t^A(0) + (1-p) Q_t^A(1) - \omega |p - p_t|$, then $p_{t+1} = \arg\max_{p \in [0,1]} f(p)$.

We know that $f(p)$ is a linear function of $p$ in both intervals $[0, p_t]$ and $[p_t, 1]$ and the maximums of linear function are always achieved in the endpoints of one interval. Thus, we have $p_{t+1} = \arg\max_{p \in \{0, p_t, 1\}} f(p)$, which means we only need to consider

$$f(0) = Q_t^A(1) - \omega p_t$$
$$f(1) = Q_t^A(0) - \omega(1 - p_t)$$
$$f(p_t) = Q_t^A(1) + p_t(Q_t^A(0) - Q_t^A(1)).$$

Next, we can build a matrix game with the property $b = \max\{a, b, c, d\} > c > d > 0 > a$. In this case, $M = 2\|Q\|_\infty = 2b$ and $\omega = \frac{(N-1)M}{N} = b$. Then we consider the condition $f(0) > f(p_t)$. We have

$$f(0) - f(p_t) = -p_t \left(Q_t^A(0) - Q_t^A(1) + \omega\right) = -p_t \left(2b - d - (b + c - a - d)q_t\right)$$
$$\Rightarrow f(0) > f(p_t) \quad \Leftrightarrow \quad q_t > \frac{2b - d}{b + c - a - d} \triangleq \tilde{q}.$$

We need $\tilde{q} < 1$ to ensure a feasible $q_t$ can be found, which means $b < c - a$.

Thus, for a matrix game satisfying the condition $c - a > b = \max\{a, b, c, d\} > c > d > 0 > a$, we can find a non-trivial solution to (13). To empirically verify this conclusion, we choose a matrix game with $(a, b, c, d) = (-4, 7, 6, 4)$ where $\tilde{q} = \frac{10}{13} \approx 0.769...$. For simplicity, we call this matrix game as matrix game case 4. We also choose $(p_0, q_0) = (0.55, 0.8)$ to ensure the condition $q_t > \tilde{q}$. The empirical results are illustrated in Figure 14. We can find the non-trivial update for the joint policy which verifies our conclusion discussed before.

### F.5 Discussions about using global state $s$ in theoretical results.

Using the global state $s$ for theoretical analysis has been a common practice in the study of multi-agent reinforcement learning, especially in the setting of decentralized learning. There are many previous works containing theoretical results in decentralized learning, which include both cooperative settings (Jiang & Lu, 2022) and non-cooperative settings (Arslan & Yüksel, 2016; Mao et al., 2022a; Zhang et al., 2024). The

main reason for this common practice is the difficulty in solving a POMDP, which has been studied for decades in Papadimitriou & Tsitsiklis (1987); Mundhenk et al. (2000); Vlassis et al. (2012). Additionally, the theoretical analysis of Dec-POMDP will be even more difficult in the multi-agent setting. If we include partial observability in the analysis, we may not obtain anything since the problem may be undecidable in Dec-POMDP (Madani et al., 1999).

