# OpenReview forum: "$f$-Divergence Policy Optimization in Fully Decentralized Cooperative MARL"
_TMLR — Accepted by TMLR_

### Review · Reviewer_LEMY · 2025-01-24

**Summary Of Contributions:**

This paper provides a new independent policy optimization formulation for cooperative MARL, f-divergence policy optimization. Further, a fully decentralized independent learning algorithm, TVPO, is provided along with detailed theoretical and empirical analysis. The empirical analysis covers three cooperative testbeds and a suite of strong baselines, where the advantages of TVPO are demonstrated. This paper's contributions are focussed on the much harder and more general decentralized training and execution setting (as opposed to several centralized training techniques in the literature).

**Audience:**

Yes

**Claims And Evidence:**

Yes

**Requested Changes:**

I request the authors to address the comments in the Weaknesses listed above.

This is a good paper. I am happy to recommend acceptance of the paper and the changes requested are not critical to securing my recommendation. I am simply providing these comments for improvement.

**Strengths And Weaknesses:**

Strengths:

The paper is well organized and motivated. It takes on the challenge of providing a theoretically principled fully-decentralized independent-learning framework and algorithm for cooperative MARL, whose difficulty is well-acknowledged in the literature. The paper has strong results with clear evidences for their major claims both theoretically and empirically.

Weaknesses:

The exposition of the work requires some polishing. I am listing the major problems here and I hope that the authors can address these points for improving their work.

1. The paper mentions that they are providing a "more general" policy optimization formulation in several places. This is mentioned right in the abstract without any basis which makes it confusing. I am not clear what this is being compared to for the claim that this work is more general. In the introduction it is mentioned that the comparison is to DPO, but only a limitation of DPO is noted (and it is not immediately clear how this paper addresses this limitation).

2. Using the agent index as a subscript and a superscript is confusing. For example, in Section 3, the agent index is used as a superscript for the policy and a subscript for the action. I suggest that the authors pick one and be consistent.

3. There are new definitions introduced at different parts of the paper without a coherent structure. I suggest the authors to use the Definition environment in Latex for each definition (especially the definitions in Page 5 associated with Eq.s 7 and 8).

4. There are certain unclear phrasings in the paper. For example, in page 6, "definition is well-defined" (what does that mean?), page 6, "critical bridge between value functions" (how?), and "we usually use Gaussian distribution" (when and what other distributions are used),  also, in Page 9, "useful properties like Appendix F.4" (what properties are referred to?).

5. Several experimental results are close and the authors should perform a statistical significance analysis. Particularly, the results in Fig. 7, Fig 8, Fig. 4(a) requires statistical significance discussions.

6. I find the arguments in the later part of Section 5.1 quite hard to follow. The conclusion is that the empirical results agree with the theoretical derivation on all four iterations. The paper must elucidate more as to which results are being demonstrated here and how the conclusions are inferred.

 7. The placement of Figure 7 seems to misaligned with the rest of the text, with the description placed along the same paragraph as the main text. This needs to be fixed.


Minors:

Page 2: allowedto -> allowed to
Page 4: such IQL -> such as IQL
Page 8: the experiment contains -> the experiments contain
Page 11: each agent will control -> each agent controls

---

> ### Author Response · Authors · 2025-02-15
>
> > The paper mentions that they are providing a "more general" policy optimization formulation in several places. This is mentioned right in the abstract without any basis which makes it confusing. I am not clear what this is being compared to for the claim that this work is more general. In the introduction it is mentioned that the comparison is to DPO, but only a limitation of DPO is noted (and it is not immediately clear how this paper addresses this limitation).
>
> Thank you for your comments. With the description "more general", we would like to present that $f$-divergence policy optimization is a broader class compared with the fixed formulation of DPO, because we have many choices for the $f$-divergence. We think the flexibility of this formulation has the potential to solve the fully decentralized learning, at least we have found total variation distance and Hellinger distance with convergence guarantees in this formulation. We will add these explanations in the revision.
>
> As for the comparison between TVPO and DPO, we present some discussions of comparison in Section 4.3 and Section 5.2. From the theoretical perspective, TVPO provides a better relation of the joint policy objective and individual policy objective, which may introduce less biases and be less possible to be trapped in the trivial policy update. From the empirical perspective, TVPO will not be trapped in some matrix games where DPO will and performs better than DPO in other more complex environment.
>
> > 1. Using the agent index as a subscript and a superscript is confusing. For example, in Section 3, the agent index is used as a superscript for the policy and a subscript for the action. I suggest that the authors pick one and be consistent.
> > 2. There are new definitions introduced at different parts of the paper without a coherent structure. I suggest the authors to use the Definition environment in Latex for each definition (especially the definitions in Page 5 associated with Eq.s 7 and 8).
> > 3. The placement of Figure 7 seems to misaligned with the rest of the text, with the description placed along the same paragraph as the main text. This needs to be fixed.
> > 4. Page 2: allowedto -> allowed to Page 4: such IQL -> such as IQL Page 8: the experiment contains -> the experiments contain Page 11: each agent will control -> each agent controls
>
> Thank you for your suggestions. We will correct these contents in the revision.
>
> > There are certain unclear phrasings in the paper. For example, in page 6, "definition is well-defined" (what does that mean?), page 6, "critical bridge between value functions" (how?), and "we usually use Gaussian distribution" (when and what other distributions are used), also, in Page 9, "useful properties like Appendix F.4" (what properties are referred to?).
>
> For "definition is well-defined", the equation 7 and 8 are fixed point equations, which may have no solution or more than one solutions. We use $\gamma$-contraction to prove the corresponding fixed point equations have and only have one solution, which is the V-function we defined. Therefore the definition is well-defined.
>
> For "critical bridge between value functions", the LHS of (10) corresponds to $V^{\boldsymbol{\pi}}(s)$ and the first term of RHS of (10) corresponds to $V^{\boldsymbol{\pi}}_{\boldsymbol{\rho}}(s)$.
>
> For "we usually use Gaussian distribution", it is a common practice to use Gaussian distribution for the continuous action space and other distributions have been hardly used, to the best of our knowledge.
>
> For "useful properties like Appendix F.4", with the useful property that the maximum will only be achieved in the endpoints of an interval, we can obtain the exact solution for the TV-iteration.

---

> ### Author Response · Authors · 2025-02-15
>
> > Several experimental results are close and the authors should perform a statistical significance analysis. Particularly, the results in Fig. 7, Fig 8, Fig. 4(a) requires statistical significance discussions.
>
> Thank you for your advice. We have mentioned in Section 5 that all the learning curves correspond to five different random seeds and the shaded area corresponds to the 95% confidence interval. Therefore, we think the shaded area in these figures can reflect the statistical significance to some extent. Moreover, for your concern of statistical significance discussions, we conduct ANOVA tests over the results in Figure 4(a), 7 and 8.
>
> |     | Figure 4(a) | Figure 7 | Figure 8(1-1) | Figure 8(1-2) | Figure 8(1-3) | Figure 8(2-1) | Figure 8(2-2) | Figure 8(2-3) |
> | --- | --- | --- | --- | --- | --- | --- | --- | --- |
> | F-score | 5.651 | 12.789 | 14.997 | 7.517 | 2.842 | 3.745 | 15.336 | 14.702 |
> | p-value | **5.59e-03** | **9.97e-05** | **2.33e-05** | **2.042e-03** | 7.264e-02 | **3.251e-02** | **1.951e-05** | **2.720e-05** |
>
> Here Figure 8(1-2) means the subfigure in the first row and second column of Figure 8. We could find that all the results have $p<0.05$ and can be considered statistically significant except the results in Figure 8(1-3), which shows $\delta$ has less impact on the performance when $\alpha=3$ in the ablation study.
>
> > I find the arguments in the later part of Section 5.1 quite hard to follow. The conclusion is that the empirical results agree with the theoretical derivation on all four iterations. The paper must elucidate more as to which results are being demonstrated here and how the conclusions are inferred.
>
> In Section 4.1, we show that for all $f$-divergences, with an appropriately constructed matrix game, the changes of two agents' policies $\{p_t\}$ and $\{q_t \}$ will be determined by thresholds $\hat{p}$ and $\hat{q}$. The thresholds are determined by the matrix game (the values of $a,b,c,d$). With this property, we can make sure $\{p_t\}$ and $\{q_t \}$ converge to suboptimal solutions by choosing the initial policies $p_0$ and $q_0$.
>
> In Section 5.1, we select four different $f$-divergences, which demonstrate the property in Section 4.1 applies for all $f$-divergences, and select four different initial policies, which demonstrate the influence of the thresholds $\hat{p}$ and $\hat{q}$. From the discussions in Section 4.1, the inital policies init_1 and init_3 satisfy the condition where the converged policies will be suboptimal. The empirical results agree with this conclusion and also show that $f$-divergence will fall into sub-optimal solutions in some matrix games.

---

### Review · Reviewer_tDXb · 2025-01-31

**Summary Of Contributions:**

This paper contributes with a general f-divergence policy optimization framework for fully decentralized cooperative MARL, and an algorithm within the framework based on total variation distance. The algorithm is proven to converge to the optimal solution. Additionally, the algorithm outperforms other fully decentralized baselines in several relevant environments.

**Audience:**

Yes

**Claims And Evidence:**

Yes

**Requested Changes:**

The most relevant clarifications in order are, in my view, the following:
- How, being an instance of f-divergence, the algorithm TVPO based on total variation distance, being a particular instance of the f-divergence framework, is able to circumvent the issues exposed earlier;
- What is meant by "trivial updates" throughout the paper;
- Why is it necessary to introduce different definitions of Q and V functions to propose the TVPO algorithm, and what are the implications of solving for these different definitions rather than the original.

Less relevant issues are:
- In the introduction, it is mentioned that there are experiences in SMAC V2, which don't appear in the Experiments section;
- "allowedto" should be "allowed to" in the Related Work section

**Strengths And Weaknesses:**

Strengths:
- The problem of fully decentralized policy optimization in MARL is relevant. Particularly, the theoretical problem of convergence is especially relevant and this paper provides a good contribution.
- The algorithm proposed clearly outperforms the baselines in the experiments presented.

Weaknesses:
- It is unclear how the paper can show at the same time that (i) in general f-divergence policy optimization in fully decentralized cooperative MARL is limited and may converge to a sub-optimal solution but that (ii) a particular instance of the framework the authors propose, based on total variation, is able to circumvent the limitation.
- The paper focuses on fully observable settings in MARL, which limits the applicability of the findings.

---

> ### Author Response · Authors · 2025-02-14
>
> > It is unclear how the paper can show at the same time that (i) in general f-divergence policy optimization in fully decentralized cooperative MARL is limited and may converge to a sub-optimal solution but that (ii) a particular instance of the framework the authors propose, based on total variation, is able to circumvent the limitation.
>
> Thank you for your comments. We need to make some clarifications for our theoretical results, which can be divided into two parts. The first part, in Section 4.1, shows that all f-divergence policy optimizations will converge to suboptimal solutions in **some** cooperative Markov games. The second part, in Section 4.2, shows that TVPO, a particular instance with the total variation distance, will converge in fully decentralized learning for **all** cooperative Markov game (though it may still converge to suboptimal solution in **some** Markov games according to the first part). Other types of f-divergence may not have the similar property, to the best of our knowledge.
>
> > The paper focuses on fully observable settings in MARL, which limits the applicability of the findings.
>
> We provide some discussions about using global states in Appendix F.5. In conclusion, the global state assumption is only for theoretical analysis, where introducing partial observation is almost impossible proven by previous work. Moreover, our empirical results show the effectiveness of TVPO in partial observable environments such as SMAC.
>
> > What is meant by "trivial updates" throughout the paper;
>
> The trivial updates in the policy iteration means the new policy stay the same as the old policy after the policy iteration. If $\omega$ is too large, then for any old policy, the policy iteration will make no changes and be meaningless. Examples can be found in Figure 2 and Figure 3.
>
> > Why is it necessary to introduce different definitions of Q and V functions to propose the TVPO algorithm, and what are the implications of solving for these different definitions rather than the original.
>
> Introducing different definitions of Q and V functions serves for the proof of Theorem 4.6. Intuitively, we would like to find a relation between the joint policy objective (LHS of (10)) and the decentralized policy objective (first term of RHS of (10)). We found that replacing the $\pi^{-i}_{\operatorname{old}}$ with a fixed $\rho^{-i}$, which is a more general case, can reduce the complexity in the formulation and reveal some reveal useful properties like Proposition 4.5. Proposition 4.5 is an important part in the monotonic improvement property proof of Theorem 4.6. Moreover, we hope these different definitions can provide some insights for related researchers.
>
> > In the introduction, it is mentioned that there are experiences in SMAC V2, which don't appear in the Experiments section;
>
> The experiments of SMAC v2 are included in Figure 6, 7 and 8 on page 11.
>
> > "allowedto" should be "allowed to" in the Related Work section
>
> Thank you for pointing out our typos in this paper. We will correct these statements in the revision.

---

> > ### Comment · Reviewer_tDXb · 2025-03-14
> > **Response to rebuttal**
> >
> > I thank the authors for their response. My questions about the theoretical results were clearly explained. I also reviewed the results for SMAC v2 (I still do not know how I could miss them), and in my view they further confirm the efficacy of TVPO compared to the baselines (most importantly against DPO).

---

### Review · Reviewer_kZ6m · 2025-02-10

**Summary Of Contributions:**

The paper consider the multi-agent setting where global state is observable and agents are not allowed to share parameter weights. It proposes a new algorithm named TVPO, which uses total variance distance rather than the commonly used KL-divergence to restrict the consecutive policy update. They show that this algorithm is convergence and can avoid the suboptimality issue induced by the original KL-regularized algorithm.

**Audience:**

Yes

**Broader Impact Concerns:**

Not applicable.

**Claims And Evidence:**

Yes

**Requested Changes:**

* A rigorous justification on the assumption of global state is essential to the impact of this work.
* The presentation should be improved for readers to understand the main contribution of the paper. In particular, the paper should include more discussions on the insight of why using total variation distance. The main technical difficulties for analyzing the algorithm's convergence over the baseline methods (e.g. DPO) should be discussed as well.

**Strengths And Weaknesses:**

**Strengths**:
* The experiment results show that the proposed TVPO exhibits faster reward convergence compared to the selected baselines. The algorithm's performance is stable across different hyperparameter choices, according to the ablation results.
* The proposed algorithm is shown to be convergent and can avoid the suboptimal minimizer of the constructed example.

**Weaknesses**:
- The assumption that the global state is accessible to all agents is not practical. Under this assumption, it would be feasible to directly apply centralized algorithms, which weaken the impact of this work.
- It would be helpful if the author can highlight the insight on why using the total variation distance can resolve the suboptimality issue.
- There are some typos and missing/incorrect definitions, making the paper hard to follow. Some examples:
    - In Equation (3), the expectation should be taken w.r.t. $a^{-i}$ rather than $\pi^{-i}$.
    - In Equation (4), $s$ and $\pi_{\text{old}}$ are not defined.

---

> ### Author Response · Authors · 2025-02-14
>
> > The assumption that the global state is accessible to all agents is not practical. Under this assumption, it would be feasible to directly apply centralized algorithms, which weaken the impact of this work.
>
> As our discussions in Appendix F.5, the global state assumption is only for theoretical analysis, where introducing partial observation is almost impossible proven by previous work. Our empirical results show the effectiveness of TVPO in partial observable environment such as SMAC.
>
> Moreover, even with the global state, agents can not observe actions of other agents in fully decentralized learning. The setting of fully decentralized learning requires decentralized control or execution, which means centralized algorithms are not appropriate for this problem.
>
> > It would be helpful if the author can highlight the insight on why using the total variation distance can resolve the suboptimality issue.
>
> We need to clarify that total variation distance can not resolve the suboptimality issue. The suboptimality issue applies for all f-divergences as our discussions in Section 4.1.
>
> The insight of using the total variation distance is for the proof of the inequality (10) in Lemma 4.4. We can bridge the gap between the joint policy objective (LHS of (10)) and the decentralized policy objective (first term of RHS of (10)) with the total variation distance. Then we can prove the convergence guarantee in Theorem 4.6.
>
> In conclusion, with the total variation distance, the decentralized policy iteration will converge in fully decentralized learning (though it may still converge to suboptimal solution in some Markov games according to Section 4.1). Other types of f-divergence may not have the similar property, to the best of our knowledge.
>
> > There are some typos and missing/incorrect definitions, making the paper hard to follow.
>
> Thank you for pointing out our typos in this paper. In equation (3), the expectation should be over $a^{-i} \sim \pi^{-i}(\cdot | s^\prime)$. $s$ and $\pi^i_{\operatorname{old}}$ can be any fixed ones in equation (4). We will correct these statements in the revision.
>
> > The main technical difficulties for analyzing the algorithm's convergence over the baseline methods (e.g. DPO) should be discussed as well.
>
> The main technical difficulties is the gap between the joint policy objective (LHS of (10)), which is needed for the proof of the convergence, and the decentralized policy objective (first term of RHS of (10)), which is the objective can be optimized in fully decentralized learning. If we can find an appropriate relation such as inequality (10), then the convergence can be proved following the classic path of the monotonic improvement property like our proof of Theorem 4.6.
>
> We present some discussions of comparison between TVPO and DPO in Section 4.3 and Section 5.2. From the theoretical perspective, TVPO provides a better relation which may introduce less biases and be less possible to be trapped in the trivial policy update.

---

### Review · Reviewer_G6n7 · 2025-02-12

**Summary Of Contributions:**

The paper presents a new algorithm to solve collaborative decentralised multi-agent learning. The algorithm is based on the idea of using the f-divergence, concretely proposing total variation policy optimisation.

**Audience:**

Yes

**Claims And Evidence:**

Yes

**Requested Changes:**

See above

**Strengths And Weaknesses:**

I have some confusion about the contributions of this paper. I'll state them in a series of questions, hoping that the authors will help me to clarify the matter.

1. Is the main contribution of the paper theoretical or empirical?
2. Is the f-divergence formula (4) something new? In particular, why not express this in terms of a single policy outputting all actions?
3. Following, the derivations in Sec 4.1 feel standard RL calculations. Is it new, or at which point?
4. How does the fixed point defined in Thm 4.6 depend on $\omega$? Could you give some examples where dependence is not trivial?
5. Do the results of Sec 4.1 hold for any f-divergence (or a broader class than TV used in Thm 4.6). In particular, is TV beneficial from the theoretical standpoint, or only empirically better (this is perfectly fine; I just want this to be clear)?
6. Is it correct to say that the experiments with SMAC do not fully fall into the framework developed in Sec 4.1. That is, SMAC is not fully observable and states are agent dependant, while Sec 4.1 assumes the full Markov state?

---

> ### Author Response · Authors · 2025-02-14
>
> > Is the main contribution of the paper theoretical or empirical?
>
> Our paper makes **both theoretical and empirical contributions**. Theoretically, it introduces the $f$-divergence policy optimization framework for fully decentralized MARL, analyzes its limitations, and formally proves the convergence of TVPO (a specific instance using total variation divergence). Empirically, it demonstrates TVPO’s superiority over state-of-the-art baselines across multiple benchmarks (SMAC, MuJoCo, SMACv2) and validates the theoretical limitations of $f$-divergence formulations via matrix games.
>
> > Is the f-divergence formula (4) something new? In particular, why not express this in terms of a single policy outputting all actions?
>
> The $f$-divergence formulation (Equation 4) is **novel in the context of fully decentralized MARL**. While similar policy regularization ideas exist in single-agent RL (e.g., KL-divergence in TRPO/PPO), this work explicitly generalizes to $f$-divergences and adapts it to independent learning in MARL.
>
> As our paper focuses on the fully decentralized MARL, the formulation is decentralized by design: each agent independently optimizes its own policy (Equation 4) without assuming access to other agents’ policies, aligning with the strict DTDE setting (no communication or parameter sharing). A joint policy formulation would violate decentralization requirements.
>
> > Following, the derivations in Sec 4.1 feel standard RL calculations. Is it new, or at which point?
>
> The formulation in Lemma 4.1 is obtained by following the idea of [1] in single-agent RL as we mentioned in Section 4.1. However, the key novelty of Section 4.1 lies in Proposition 4.2 and Corollary 4.3, which demonstrate independent updates can converge to suboptimal equilibria via a matrix game that even with f-divergence regularization. This highlights a fundamental limitation of the f-divergence framework in cooperative MARL, which is not apparent in single-agent RL.
>
> [1] Wenhao Yang, Xiang Li, and Zhihua Zhang. A regularized approach to sparse optimal policy in reinforcement
> learning. In Advances in Neural Information Processing Systems (NeurIPS), 2019.
>
> > How does the fixed point defined in Thm 4.6 depend on $\omega$ ? Could you give some examples where dependence is not trivial?
>
> The dependence lies in Lemma 4.4. We prove inequality (10) for $\omega = \frac{(N-1)L}{N}$, which results in the monotonic improvement property and the convergence guarantee in Theorem 4.6. The inequality (10) is a non-trivial relation between the joint policy objective (LHS of (10)) and the decentralized policy objective (first term of RHS of (10)). Intuitively, a smaller $\omega$ means less biases, however, this $\omega$ is the best result we can obtain for inequality (10).
>
> > Do the results of Sec 4.1 hold for any f-divergence (or a broader class than TV used in Thm 4.6). In particular, is TV beneficial from the theoretical standpoint, or only empirically better (this is perfectly fine; I just want this to be clear)?
>
> Yes, the results of Sec 4.1 hold for any f-divergence. Figure 1 and Section 5.1 show the phenomenon we discussed in Section 4.1 by empirical results for four types of f-divergence.
>
> TV is beneficial from the theoretical standpoint as we prove Lemma 4.4 and Theorem 4.6 with TV. Therefore, the policy iteration with TV has a convergence guarantee for fully decentralized learning.
>
> > Is it correct to say that the experiments with SMAC do not fully fall into the framework developed in Sec 4.1. That is, SMAC is not fully observable and states are agent dependant, while Sec 4.1 assumes the full Markov state?
>
> Yes. Previous study has shown the difficulty for finding a solution in partial observable environment (more detailed discussions are included in Appendix F.5). So we follow the common practice to use state $s$ in theoretical analysis.

---

### Decision · Action_Editor_SEUW · 2025-03-24

**Recommendation:** Accept as is

**Comment:**

This paper is correct overall. It is easy to implement and could have practical use cases, as supported by the experiments and ablation studies.

**Audience:**

Yes.

**Claims And Evidence:**

Yes, with sufficient experiments and ablation studies.